# Altered developmental programs and oriented cell divisions lead to bulky bones during salamander limb regeneration

Marketa Kaucka [1,17], Alberto Joven Araus [2,17], Marketa Tesarova [3], Joshua D. Currie[4], Johan Boström[5], Michaela Kavkova [3], Julian Petersen[5,6], Zeyu Yao[2], Anass Bouchnita[7,8], Andreas Hellander[7], Tomas Zikmund [3], Ahmed Elewa[2,9], Phillip T. Newton [10,11], Ji-Feng Fei [12,13], Andrei S. Chagin [14,15], Kaj Fried[16], Elly M. Tanaka [12], Jozef Kaiser [3], András Simon [2] ✉ & Igor Adameyko [5,14] ✉

There are major differences in duration and scale at which limb development and regeneration proceed, raising the question to what extent regeneration is a recapitulation of development. We address this by analyzing skeletal elements using a combination of micro-CT imaging, molecular profiling and clonal cell tracing. We find that, in contrast to development, regenerative skeletal growth is accomplished based entirely on cartilage expansion prior to ossification, not limiting the transversal cartilage expansion and resulting in bulkier skeletal parts. The oriented extension of salamander cartilage and bone appear similar to the development of basicranial synchondroses in mammals, as we found no evidence for cartilage stem cell niches or growth plate-like structures during neither development nor regeneration. Both regenerative and developmental ossification in salamanders start from the cortical bone and proceeds inwards, showing the diversity of schemes for the synchrony of cortical and endochondral ossification among vertebrates.

Salamanders (such as newts and the axolotl) are the only tetrapods that regenerate entire limbs during their lifespan[1]. Limb regeneration starts with scarless wound healing and proceeds with the subsequent formation of a blastema that gives rise to lost limb elements distal to the plane of amputation[2,3]. Single-cell sequencing data in axolotl showed that a majority of blastema cells are funneled through a transient embryonic limb bud-like state before re-differentiation to various tissues[1,4]. Although this transient cell state indicates that limb regeneration is a recapitulation of embryonic development, regeneration has distinctive features. One such process is reversing the terminally

[1]Max Planck Institute for Evolutionary Biology, Plön 24306, Germany. [2]Department of Cell and Molecular Biology, Biomedicum, Karolinska Institute, Stockholm 17165, Sweden. [3]Central European Institute of Technology, Brno University of Technology, Brno 61200, Czech Republic. [4]Department of Biology, Wake Forest University, Winston-Salem, NC, USA. [5]Department of Neuroimmunology, Center for Brain Research, Medical University Vienna, Vienna 1090, Austria. [6]Department of Orthodontics, University of Leipzig Medical Center, Leipzig 04103, Germany. [7]Department of Information Technology, Uppsala University, Uppsala, Sweden. [8]Department of Mathematical Sciences, The University of Texas at El Paso, El Paso, TX 79902, USA. [9]Department of Genetics, Microbiology and Statistics, Faculty of Biology, University of Barcelona, Barcelona, Spain. [10]Department of Women's and Children's Health, Karolinska Institute, Solna, Sweden. [11]Astrid Lindgren Children's Hospital, Karolinska University Hospital, Solna, Sweden. [12]The Research Institute of Molecular Pathology (IMP), Vienna 1030, Austria. [13]Department of Pathology, Guangdong Provincial People's Hospital, Guangdong Academy of Medical Sciences, Guangzhou 510080, China. [14]Department of Physiology and Pharmacology, Karolinska Institutet, Stockholm 17165, Sweden. [15]Department of Internal Medicine and Clinical Nutrition, University of Gothenburg, Göteborg 41346, Sweden. [16]Department of Neuroscience, Biomedicum, Karolinska Institute, Stockholm 17165, Sweden. [17]These authors contributed equally: Marketa Kaucka, Alberto Joven Araus. ✉e-mail: andras.simon@ki.se; igor.adameyko@ki.se

differentiated state in muscle fibers, an essential step during blastema formation in newts[5]. Another example of a regeneration-specific process is the nerve-dependent proliferation of blastema cells. If limbs are denervated, blastema growth and consequent regeneration are halted[6]. Interestingly, mouse digit tip regeneration may also be nerve-dependent, indicating an evolutionarily conserved mechanism behind the re-growth of a mammalian body part[7–10].

Another difference between development and regeneration is represented by the scale at which the two processes occur with up to an order of magnitude difference in dimensions. In a regenerating limb, patterning and emergence of cell type heterogeneity occur when the structure has a much larger size, growth speed and environmental constraints compared to embryonic development. Given these constraints, it is plausible to assume that the dynamics of regenerative morphogenesis differs from embryonic development. Nevertheless, developmental mechanisms do play a major role in the patterning and differentiation of multiple lineages during regeneration, such as BMP, FGF and other signaling pathways[11–13].

Although regeneration results in a limb of the same size as the contralateral counterpart, the fidelity of regeneration is not perfect. Patterning defects have been documented in the skeletal muscle of the regenerated limb[14], and paleontological findings also suggested anomalies in skeleton regeneration[15].

The regenerating skeleton originates from several cellular sources, including mainly periskeletal cells and dermal fibroblasts[4,16,17]. Tracing experiments revealed no lineage-specific stem cells for cartilage and bone, and instead, the periskeletal cells extend the bone via generating an ossifying cartilaginous callus, whereas dermal and interstitial fibroblasts generate new distal segments of cartilage and bone[4]. The capacity of periskeletal cells to generate chondrogenic progenitors might explain the healing, recovery and regeneration of the joints and joint cartilages after damage[18].

In mammals, after the phase of primary embryonic chondrogenesis, cartilages within developing limbs undergo endochondral ossification. They expand in a longitudinal direction while converting into a bone[19,20]. The elongation of bones in a growing mammalian limb is driven by the growth plate, a tiny disc of cartilage located at each end of every long bone and morphologically subdivided into resting, proliferative and hypertrophic zones. The resting zone of the growth plate contains cartilage stem cells that are PTHrP (Parathyroid Hormone-related Protein) positive[21]. These cartilage stem cells reside in a specialized stem cell niche, which allows their renewal and generation of chondrocytes, required for the extension of the limbs during postnatal growth[22]. PTHrP+ stem cells generate first clones of transiently amplifying flattened chondrocytes with longitudinal orientation, forming continuously extending clonal columns, which thereafter undergo hypertrophy, i.e., enlargement in size[21,22]. This oriented cellular arrangement in the limb cartilage defines the directionality of the expansion of rod-shaped skeletal elements[22,23]. Subsequent chondrocyte differentiation is maintained by both the PTHrP signaling originating from undifferentiated cells in the resting zone and the release of IHH (Indian hedgehog homolog) by the hypertrophic chondrocytes[24].

The presence of stem cells and their niche, extensive cell amplification and hypertrophy underlie unidirectional limb growth[21,22,25]. However, this is not the only mechanism of endochondral ossification following the initial chondrogenesis. The extension and ossification of a cranial base during embryonic development in mammals is different and do not depend on the stem and transiently amplifying cells in basicranial synchondroses[26]. Instead, the expansion of a cranial base during ossification depends on the number of initially deposited embryonic chondrocytes that become reoriented due to convergent extension and gradually depleted during the transition to hypertrophy[23].

In this paper, we set out to compare the mechanisms of developmental and regenerative formation of the limb skeleton in salamanders. Micro-computed tomography, gene expression pattern analyses, lineage tracing and exploration of proliferation patterns revealed fundamental differences between regeneration and development of the limb skeleton. We found that chondrogenesis and osteogenesis are uncoupled in time during regeneration but not during development. Furthermore, we demonstrate the absence of cartilage stem cells and describe the dynamics of oriented cell divisions, cell repositioning and ossification during cartilage extension in both development and regeneration. Finally, our results explain how the rod-shaped cartilages grow and acquire a shape under different constraints.

## Results
### Different modes of skeletal elements shaping in development and regeneration
To address the differences in skeletogenesis, we collected developing, postmetamorphic and regenerating limbs from the newt *Pleurodeles waltl*, and reconstructed cartilage and bone structures employing phospho-tungstic acid (PTA) enhanced micro-CT according to a published protocol[23,27]. The identification of the border between soft tissues, cartilage and bone was done according to our previously published approach allowing reliable segmentation of soft and stiff tissues in vertebrates, and specifically in salamander species[23,28–31]. All samples were 3D-rendered, and subsequent segmentations of the skeletal parts were generated to analyze the regenerating structures (Fig. 1A, B and Fig. 2). In postmetamorphic *Pleurodeles waltl*, long bone regeneration starts with peeling off the periosteum and forming a cartilaginous cap ensheathing the bone stump one week after amputation (Fig. 1B, also observed in larval *Pleurodeles*, Fig. 2A). The cap formation is followed by progressive outgrowth and patterning of the missing skeletal elements (Fig. 1B).

Surprisingly, the entire sequence of patterning and outgrowth proceeded without ossification until the limb reached a size similar to the contralateral control limb (Fig. 2A, B). Thus, the ossification of regenerated limbs started around the time the final size was reached, and subsequently, the ossification process took several months until completion (Fig. 1A, B). Quantifications of ossified skeletal elements in post-metamorphic *Pleurodeles* after completion of regeneration (50 w.p.a.) showed that the regenerated ulna and radius contained more bone (Ulna: $2.37 \pm 0.48$ mm$^3$; Radius: $2.29 \pm 0.54$ mm$^3$) compared to the controls (Ulna: $1.42 \pm 0.56$ mm$^3$; Radius: $1.31 \pm 0.58$ mm$^3$) (Fig. 1C).

We next asked whether the differences in ossification pattern and formation of bulkier long bones were an intrinsic property of salamander limb regeneration (i.e., activation of a regenerative program rather than a re-activation of a developmental one) or a consequence of size constraints (i.e., a re-activation of developmental program constrained by the scale of the structure to be re-generated). To address this question, we first performed analogous analyses in developing and regenerating *Pleurodeles* larvae (for an in-depth discussion of these results, please see the Supplemental Note; Figs. 1D–H, 2, S1A, S2, S3B), followed by experiments in axolotls (Fig. S4). The axolotl, *Ambystoma mexicanum*, is a paedomorphic salamander that naturally does not go through metamorphosis, nevertheless, metamorphosis can be induced experimentally. Consistently with the observations in the adult *Pleurodeles*, we found that after amputation, a callus ensheathing the bone stump (Fig. 2A) was followed by the formation of sturdy, abnormal cartilages (Fig. 2B, S2, S3, S4A) and reduced ossification (Fig. 1D–H) in larval salamanders of both species. This difference in skeletal element volumes was measured and visualized with a 3D shape comparison test (Fig. 2C). We also examined regeneration in *Ambystoma* after artificially induced metamorphosis (Fig. S4B). We found that the regenerated skeletal elements also appeared with a bulkier shape than the normally developing skeleton (Fig. S4C, D). In *Pleurodeles*, the bulkier shape of regenerated cartilage was preserved since cortical bone was slowly allocated on the surface

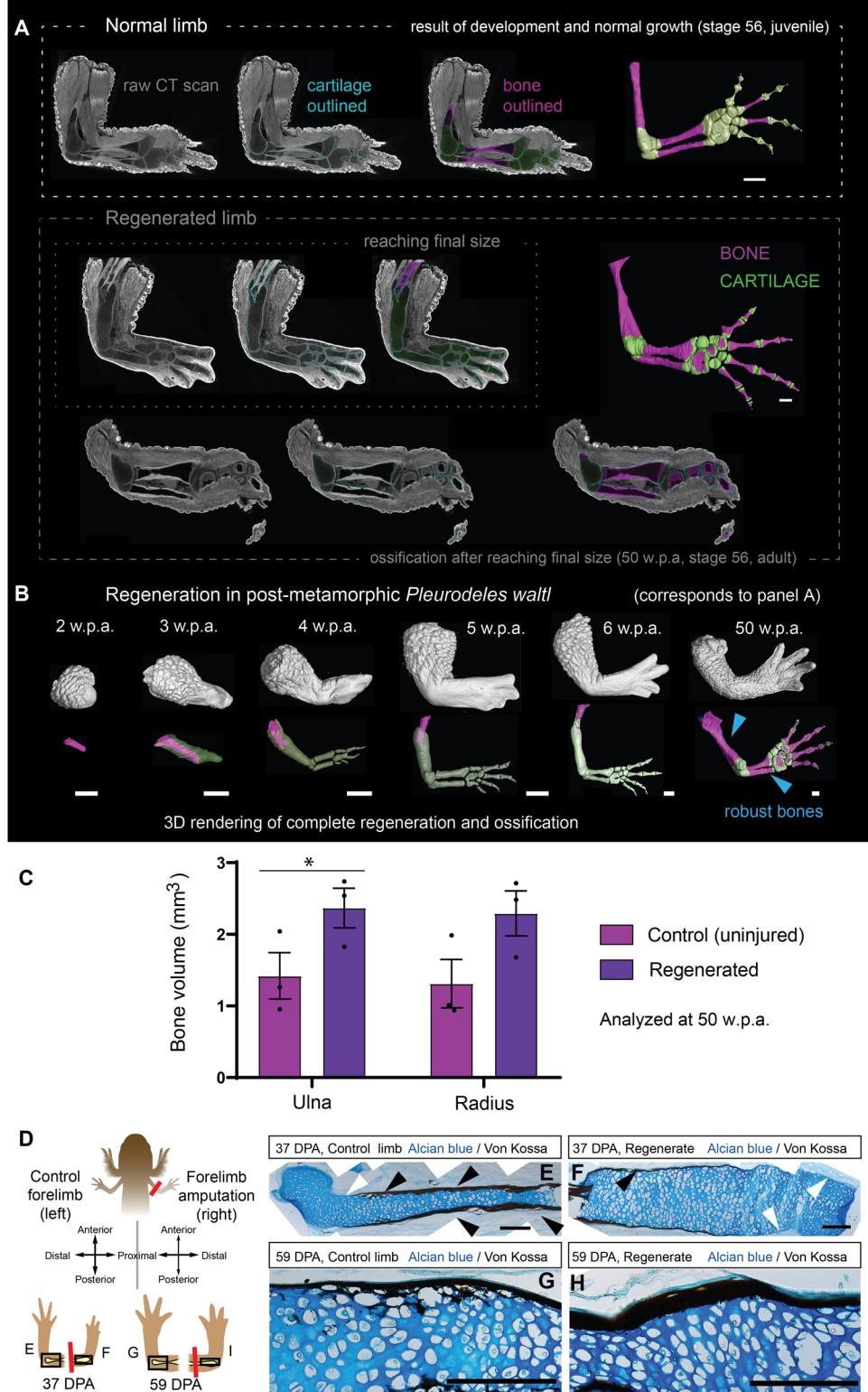

of the initially-formed cartilaginous structure without immediate erosion or substitution of the cartilage anlagen (Fig. 1B, E, F). Finally, we detected patterning defects in some of the regenerated post-metamorphic *Pleurodeles* (Fig. S1B, example 3) and *Ambystoma* samples (Fig. S4B). Overall, regenerative skeletogenesis in both larval and postmetamorphic salamanders resulted in differently shaped cartilages and bones as compared to those forming during normal development and growth.

In *Ambystoma* regenerating limbs at 32 w.p.a., we detected the ossification onset that occurred in patches: despite the skeletal elements were bulkier in the regenerated limbs, the control limbs showed a statistically non-significant larger volume of ossification (Ulna: $0.30 \pm 0.08$ mm$^3$; Radius: $0.35 \pm 0.12$ mm$^3$) than the regenerated ones (Ulna: $0.16 \pm 0.07$ mm$^3$; Radius: $0.28 \pm 0.12$ mm$^3$) (Fig. S4D), because of the initial stage of ossification in regenerating limb. Notably, ossification during limb regeneration in both salamander species started in a

**Fig. 1 | Bulky skeletal elements in the regenerating *Pleurodeles waltl* limb.**
**A** Micro-CT scans and segmented 3D models of postmetamorphic *Pleurodeles waltl* limbs with depicted slice planes showing the original CT slice (left), Avizo segmentation, i.e. non-smoothed after manual segmentation, (middle) and after smoothing, color-coded model from VG studio (right). The segmentation steps are described in detail in[60]. (upper) Uninjured limb, (middle) Regenerating limb, 5 w.p.a., (bottom) Regenerated limb, 50 w.p.a. Scale bars 1 mm. **B** Amputated limbs in post-metamorphic *Pleurodeles waltl* were analyzed at 2, 3, 4, 5, 6 and 50 weeks post-amputation (w.p.a.). Note the regenerating limb's skeletal elements consist of cartilage during the patterning and outgrowth phases of limb regeneration. Ossification did not appear until the limb reached the contralateral control's approximate size and anatomical composition. Blue arrows highlight the increased thickness found in the regenerated bones compared to contralateral control limbs. Scale bars, 1 mm. **C** Quantification of the volumes occupied by ossified bone in the radius and ulna of fully regenerated (50w.p.a.) versus control limbs of adult

*Pleurodeles waltl*. Regenerated limbs contained more volume of bone than uninjured controls (2-way ANOVA: n.s. Interaction, $p = 0.9443$; n.s. Zeugopodial element, $p = 0.7773$; * Regeneration, $P = 0.0153$). $n = 3$ limbs per condition. Data are presented as mean values +/- SEM. **D** Scheme of experimental design and location of amputation in larval *Pleurodeles waltl*. **E** Alcian blue / Von Kossa staining highlights the skeletal elements of the contralateral control humerus at 37 days post-amputation (d.p.a.). White arrow points to gap in the mineralisation, and black arrows point to the highly mineralised bone. Scale bar, 200 μm. **F** Alcian blue / Von Kossa staining highlights the skeletal elements of the regenerating humerus at 37 days post-amputation (d.p.a.). White arrows point to gaps in the mineralisation, and black arrows point to the slightly mineralised bone. Scale bar, 200 μm. **G** Alcian blue / Von Kossa staining highlights the ossification of the contralateral control humerus at 59 days post-amputation (d.p.a.). Scale bar, 100 μm. **H** Alcian blue / Von Kossa staining highlights the ossification of the regenerating humerus at 59 days post-amputation (d.p.a.). Scale bar, 100 μm.

cortical bone, and hypertrophic chondrocytes were preserved under cortical bone for a long time (Fig. 1D–H, S4, Fig. 4C).

Thus, formation of cortical bone in regenerating salamander limbs is uncoupled from cartilage growth, and corticalization of skeletal elements lags behind chondrocyte hypertrophy. This is in sharp contrast to homeostatic development of both salamander and mouse long bones, where formation of cortical bone occurs in parallel to chondrocyte hypertrophy (although Indian hedgehog (Ihh) is expressed by hypertrophic cells in all investigated cases)[32,33] (Fig. S3C, please, see Peer Review file for high-resolution images and individual channels).

## Variations in evolutionarily conserved cell dynamics define skeletal elongation and shape

We next sought to understand the cell dynamics underlying skeletal growth during development and regeneration. We performed EdU pulse-chase experiments in both salamander species, combined with clonal tracing in *Nucbow/Cytbow Pleurodeles* and *Brainbow Ambystoma*[16,34]. In order to test whether differences in spatial organization of progenitor cells could explain the differences in the thickness of cartilage elements, we took advantage of the small size and transparency of regenerating axolotl digits and performed live analysis of single cells and their progeny (Fig. 3). Re-analysis of live-imaging data[16] revealed that mesenchymal chondrogenic cells and early immature chondrocytes in the most proximal part of the digit divided predominantly along the proximo-distal axis of the limb (Fig. 3A). Later, after differentiation into chondrocytes, the orientation of cell divisions changed, allocating cells transversally (Fig. 3A). In line with this, we noticed that some perichondrally-positioned flattened cells also contributed to chondrogenic clones with similar transversal cell allocation pattern (Fig. 3B, for more examples of clonal cell arrangements in *Ambystoma*, please, see the Peer Review file). Next, we measured the orientation of EdU+ doublets in the developing and regenerating rod-shaped cartilages of both species (Fig. 4A, Fig. S5). The transversally oriented cellular arrangements (similar to those observed in live-imaging of regenerating fingertip cartilage (Fig. 3A, B)) were observed in the zeugopodial skeletal elements by EdU pulse-chase experiments in both species at various stages, both during development and regeneration (Fig. 4A, Fig. S5). Additional analysis of clonal shapes corroborated the longitudinal-to-transversal switch in cell division orientations in rod-shaped cartilages transforming into long bones in regenerating and normally developing limbs (Fig. 4B, C, Fig. S6). We could not perform live imaging on clonally traced ulna, radius or humerus due to their thickness, therefore, we analyzed clonally traced tissue sections of developing and regenerated humerus, ulna and radius to show the consistent spatial arrangements of chondrocyte clones in all skeletal elements (Fig. 4, please navigate to Peer Review file for extended clonal tracing results). We confirmed

that development and regeneration of phalange, humerus, ulna and radius employ similar cellular dynamics during growth and shaping.

Thus, in development and regeneration, cartilage anlagen in salamanders enlarges via clonal expansion during a brief phase of longitudinal and then a longer phase of transversal orientation, as well as intercalation of new clones from the perichondrium. This is in sharp contrast to growing mammalian limb where chondrogenic clones are stably oriented into longitudinal orientation before the ossification[22] (Fig. 4D). The dynamics of cell divisions during salamander skeletal regeneration and growth rather resembles the pattern observed in mammalian basicranial synchondroses (Kaucka et al., 2017) (Fig. 4E). In addition to the bone-elongation mechanisms that depend on proliferation, we discovered the elongation-related role of convergent extension processes in maturing regenerative and developing cartilage. During the pre-ossification phases of the regenerative limb skeletal growth, we observed cell flattening before chondrocyte hypertrophy (Fig. 4B, C), resembling the flattening of transiently amplifying cells in mammalian growth plates of long bones[22] (Fig. 4D). However, unlike in mammalian growth plates, this cell flattening did not correlate with clonal expansion due to cell divisions and instead coincided with the clonal arrangements spatial shift similar to convergent extension[23,35] (Fig. 4B, C). The cell flattening and related developmental dynamics reflected in clonal arrangements in cartilages of *Ambystoma mexicanum* confirmed these conclusions (Fig. S6B–D). We additionally validated the lack of clonal expansion due to cell proliferation in salamander flattening cartilage by combining Cre-activated clonal analysis with EdU pulse-chase-based tracing (Fig. S7). The convergent extension and translocation of flattening cells led to the deployment of more flat chondrocytes along the proximo-distal axis of the future bone prior to their massive enlargement during the hypertrophy phase. The resulting instant increase of chondrocytes stacked in proximo-distal vs medio-lateral direction contributed to limb elongation without additional proliferation.

A direct comparison of cell dynamics in salamanders with the convergent extension-like events in the clonally traced growing mammalian basisphenoid cartilage showed a similar cell flattening and repositioning in the proximity of hypertrophy zones in *Sox10^CreERT2^/R26^Confetti^* mice injected with tamoxifen at E12.5 and analyzed at 17.5 (Fig. 4E), which also coincided with lack of additional proliferation. This cell dynamics was radically different from clonally traced mammalian growing proximal tibia with cartilage stem cell niche and continuous proliferation of transiently amplifying cells in *Col2A1^CreERT2^/R26^Confetti^* mice injected with tamoxifen at E14.5 and analyzed at P30 (Fig. 4D). Thus, the convergent extension-based growth of cartilages in salamanders rather resembles the cell dynamics observed in mammalian basicranial synchondroses[23], instead of the stem cell niche-based unidirectional expansion in the growth plates of mammalian long bones[22].

Although the major aspects of cell dynamics in skeletal formation are similar in salamander limb development and regeneration, the

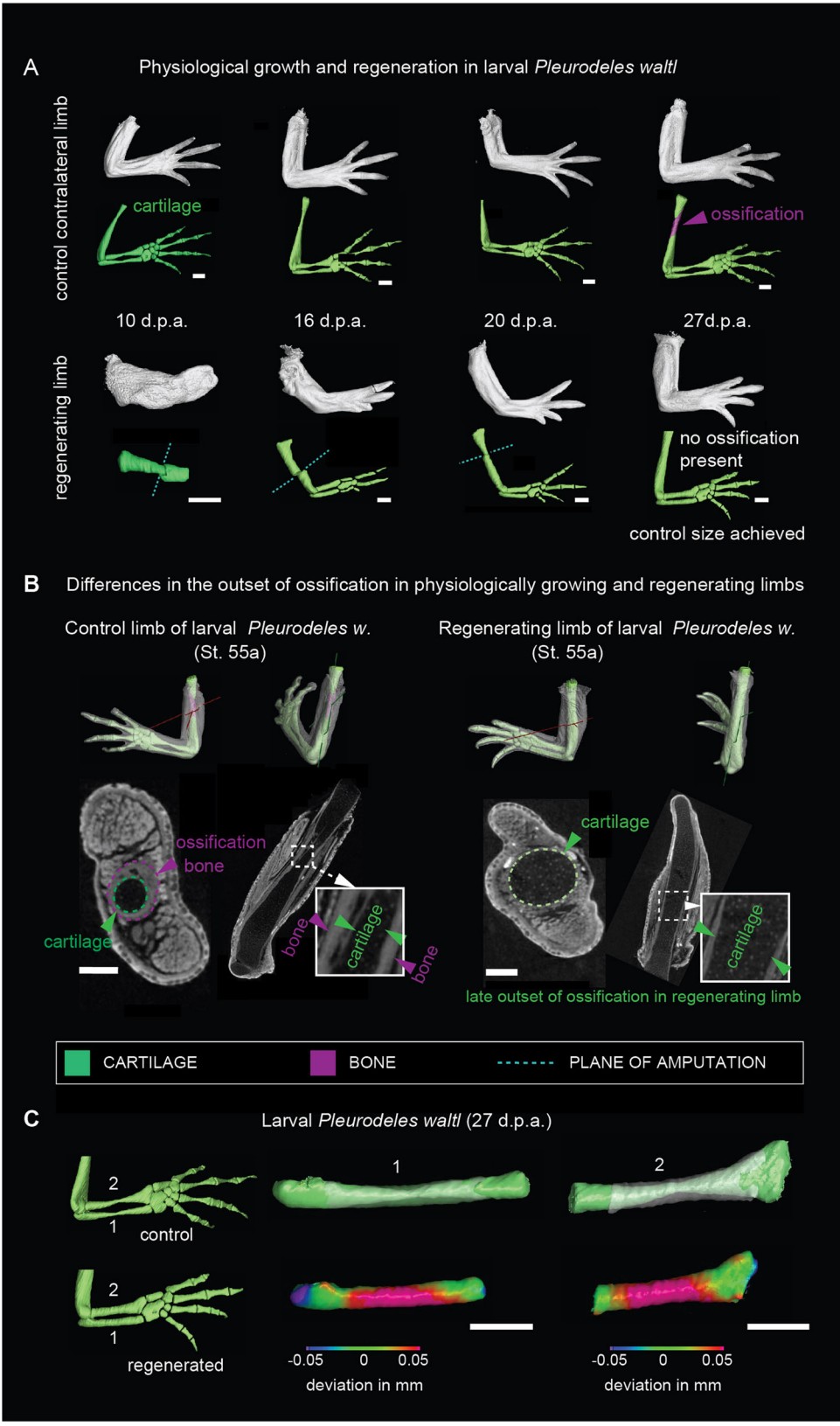

A. Physiological growth and regeneration in larval *Pleurodeles waltl*

B. Differences in the outset of ossification in physiologically growing and regenerating limbs

Control limb of larval *Pleurodeles w.* (St. 55a)

Regenerating limb of larval *Pleurodeles w.* (St. 55a)

late outset of ossification in regenerating limb

CARTILAGE　　BONE　　PLANE OF AMPUTATION

C. Larval *Pleurodeles waltl* (27 d.p.a.)

proportions of differentially oriented cell divisions may generate the observed difference in resulting skeletal elements (Figs. 1 and 2). The two phases of oriented cell divisions, longitudinal and transversal, seem to be an evolutionarily conserved mechanism during the formation and extension of the rod-shaped cartilage. The first phase leads to the cartilage elongation, whereas the second phase of transversal cell divisions leads to the increased diameter of the cylindrical rod-shaped cartilaginous element. The differences in the proportions of these two phases might lead to the corresponding differences in the resulting shapes of skeletal elements during normal development and regeneration (Figs. 1 and 2). We tested this hypothesis by mathematical modelling (Fig. 5), where we simulated the balance between

**Fig. 2 | The onset of ossification in normally developing limbs differs from regeneration. A** Micro-CT scans and 3D models of regenerating limbs in larval *Pleurodeles waltl*. Fully patterned larval limbs (stages 52a to 54) were amputated unilaterally, and limb regeneration was assessed at 10, 16, 20 and 27 days post-amputation (d.p.a.). The top panel shows physiological growth corresponding to contralateral control limbs. Note the ossification occurred in the control limbs while the regenerating limbs remained cartilaginous. Cyan dotted line point at the amputation plane. Scale bars, 500 μm. **B** Micro-CT scans and segmented 3D models with depicted slice planes showing a delayed outset of ossification in regenerating larval limb of *Pleurodeles waltl*. A representative contralateral control (left) and a regenerating limb (right) show the respective presence and absence of ossification in the humerus. Note the presence of chondrocytes underneath the ossified layer in the control limb. Green color represents the cartilage, and magenta color represents the bone. White dotted line marks the area that is magnified in the insets. Scale bars, 200 μm. **C** 3D comparisons of shapes of normally developed and regenerated skeletal elements from the forelimb of larval *Pleurodeles waltl*. Note that regenerated skeletal parts have increased diameter, the shape differences are presented as a heat-map of shape deviation. Scale bars, 500 μm.

longitudinal and transversal growth of an initially perfect rod-shaped element. Varying the proportion of longitudinal and transversal expansion in silico resulted in bulky cylindrical (regenerative) versus normal developmental shapes. The model also suggested that during development, the early cortical bone formation in the center of rod-shaped cartilage mechanically blocks further transversal cell expansion of chondrocytes and limits the thickness of the resulting bone in the middle of the normally formed skeletal part. In contrast, regenerative growth, based exclusively on cartilage expansion with delayed ossification, does not limit cartilage expansion neither in the center nor in other parts of skeletal elements and results in a different cylindrical rod-like shape.

## Cartilage regeneration and ossification involve a dynamic expression of the *PTHrP/Ihh* signaling loop

Cell proliferation, differentiation and ossification during the development and growth of the mammalian skeleton depend on the PTHrP/Ihh signaling loop[24]. Shh, Ihh and PTHrP can be seen in scRNAseq of salamander blastema cells[4], albeit the spatial location of the positive cells cannot be figured out from the datasets. To clarify this, we addressed the spatio-temporal expression patterns of *PTHrP*, *Gli1* and *Ihh* after unilateral amputations in stage 55a *Pleurodeles* larvae. We assessed the molecular patterns in regenerating (Fig. 6, Peer Review file for high-resolution images and individual channels) and contralateral control (Fig. S8, Peer Review file for high-resolution images and individual channels) humerus when the regenerates displayed similar morphological features to those shown in Fig. 1B, Fig. 2A; For this analysis, limbs were collected at 12, 20, 37, 59 and 166 d.p.a. (Fig. 6A, S8A).

In regenerating limb at 12 d.p.a., condensation of nascent cartilage was present in the core of the protruding blastema as identified by the expression of chondrocyte markers SOX9 and COL2A1 (Fig. 6B). In this region, we observed a radial arrangement of the PTHrP/Ihh loop with *Ihh* expression (Fig. 6C[1]) in the central area of the cartilage, surrounded by a *Gli1* expression domain (Fig. 6C[2]) and by *PTHrP* expression at the periphery (Fig. 6C[3]). In contrast, the contralateral control uninjured humerus showed expression domains arranged longitudinally (Fig. S8B), thereby resembling mammalian embryonic bone[24]. At 20 d.p.a., *Sox9* and *Col2a1* expression (Fig. 6D) in the regenerating skeletal elements revealed its cartilage nature, confirming micro-CT reconstructions (Fig. 2) and histological observations (Fig. S3B). Simultaneously, the shape of the growing end of the newly forming element lacked the rod-shape typical for humerus of the contralateral bones at this stage (compare the developing limb in Fig. S2D with the contralateral regenerate in Fig. 6D, E). At this timepoint, there was a conspicuous expression of *Ihh* in pre-hypertrophic chondrocytes in the proximal part of the rapidly growing humerus (Fig. 6E[1]), whereas expression of *PTHrP* and *Gli1* was constrained to the distal part (Fig. 6E[2]), indicating polarization of the *PTHrP/Ihh* loop toward the direction of longitudinal growth. We observed a few perichondrial cells expressing *Gli1/PTHrP* in the developing articulation (Fig. 6E[3]). From this stage onwards, the regenerated skeletal element appeared thicker than the contralateral control (compare Fig. 6F with S8C and Fig. 1E, F). The zone of late hypertrophic cells became morphologically noticeable at 37 d.p.a., although without any detectable *Ihh* expression (Fig. 6F[1]). The *Ihh* expression domain localised distally to the late

hypertrophic zone (likely reflecting pre-hypertrophic and early hypertrophic chondrocytes) and was smaller in size compared to 20 d.p.a. but still larger than the contralateral control (Fig. 6 as compared to Figs. S8C and 6E). At 37 d.p.a., *PTHrP*+ cells were predominantly localised at the periarticular region (Fig. 6F[4], 6F[5]), *Ihh* expressing chondrocytes retain their pre-hypertrophic localization albeit occasionally showed puncta that were positive for *Gli1* and *PTHrP* (Fig. 6F[2]). A strong expression of *Gli1* and, in some but not all limbs, *PTHrP* was found in the perichondrium (Fig. 6F[6]). By 59 d.p.a., the regenerates had nearly reached the length of the contralateral control limbs. At this stage, *Gli1* and *PTHrP* expressing cells were observed predominantly in the periskeletal cells, representing likely perichondrium (Fig. 6G[1]). *PTHrP* expression within the cartilage was less pronounced and occasionally appeared in the flat chondrocytes (Fig. 6G[2]), and *Ihh* expressing cells occupied most of the growing humerus (Fig. 6G[3]). We found *Gli1* and *PTHrP* positive puncta in the hypertrophic chondrocytes at this stage (Fig. 6G[4]). We observed a similar pattern in the contralateral limbs, suggesting that this pattern reflects the normal way of limb growth (Fig. S8B–E). The analysis of the regenerating limbs at 166 d.p.a. showed that ossification occurred in patches all along the humerus (see white arrowheads in Fig. 6H), together with a reduction in the number of cells expressing *PTHrP, Gli1 and Ihh*.

Taken together, during early stages of regenerating cartilage, the PTHrP/Ihh system is arranged radially, but thereafter repositions in proximo-distal direction and maintained in such spatial arrangement until the full size of the regenerate is achieved. Of interest, some *Ihh* expressing cells showed positivity for *Gli1* and *PTHrP*, an observation which requires further investigation. PTHrP/Ihh loop has a dynamic expression pattern during regeneration, in contrast to the gradient maintained during developmental growth in salamanders. The continuous endochondral ossification during development shows a moderate expression of *PTHrP* in the proliferative zone, followed by *Gli1*+ cells, while *Ihh* is expressed in the chondrocytic cells (Fig. S3C). In contrast, PTHrP/Ihh loop has dynamic expression domains during regeneration that vary in size and shape compared to their contralateral control limbs (Fig. 6, Fig. S8). First, cartilage condensation is characterized by a switch to a radial polarity pattern of *Ihh-Gli1-PTHrP* from the core to the periphery (Fig. 6B, C). Second, *Ihh*+ pre-hypertrophic chondrocyte domain expands and forms the callus wrapping the amputated bone, while the growth continues distally, characterized by *Gli1*+ columnar chondrocytes and *PTHrP*+ articular chondrocytes (Fig. 6D, E). Third, the gradient observed in development is restored but with extended expression domains compared to the unamputated controls (Fig. 6F). Fourth, *Gli1* and *PTHrP* double-positive enlarged, likely hypertrophic chondrocytes, are located proximally to the flat cells (Fig. 6G[4]), both in developing and regenerating limbs. We also found periskeletal cells positive for *Gli1* and *PTHrP*, which could contribute to the developing and regenerating cartilage (Fig. 6C4, E3, F6, G1, S8B[4]).

## Discussion

We identified regeneration-specific features of limb skeleton growth in salamanders. Regeneration and development of a limb are initiated under dramatically different starting conditions: a severe wound leading to the formation of a regeneration blastema versus an embryonic limb

Live imaging of regenerating digits in CAGGs::ERT-Cre-ERT-T2A-GFPnls/Brainbow axolotl

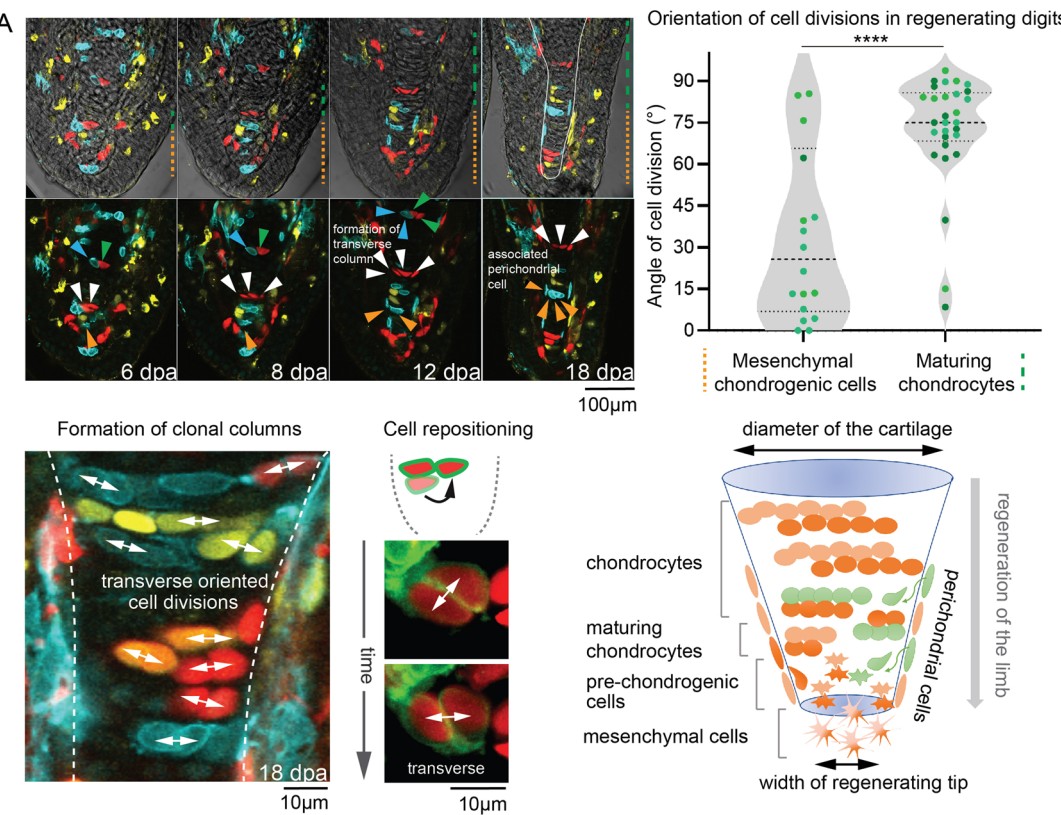

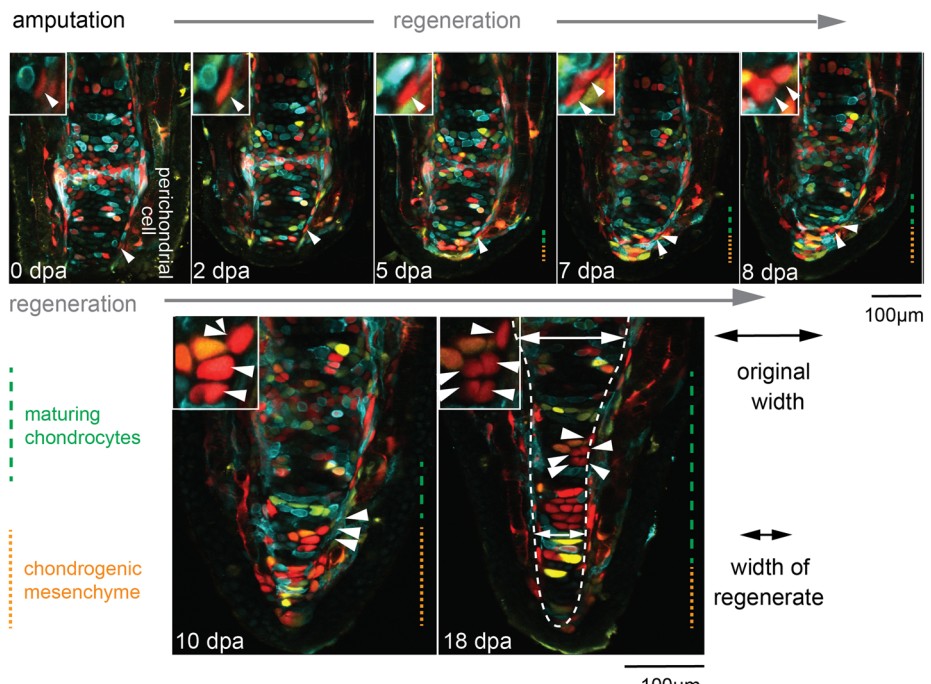

bud which is much smaller. The two different starting points influence the initiation of skeletogenesis in terms of cell sources and cell dynamics. For instance, embryonic development of skeletal elements is initiated within the limb bud from emerging chondrogenic mesenchyme[36]. The origin of regenerative skeletal elements is rather different, although the cell states in the blastema resemble the state of developing limb bud cells[4]. Unlike in development, the early stages of a salamander limb regeneration involve periosteum peeling and the appearance of a cartilaginous sheath around the severed bone, sharing features with the formation of a cartilaginous callus during mammalian bone fracture healing[37,38]. Although tail regeneration after autotomy in lizards fundamentally differs from blastema-mediated epimorphic

**Fig. 3 | Cell dynamics during skeletal elongation in *Ambystoma mexicanum*.**
**A** Live imaging of cell dynamics in the regenerating digits of genetically traced regenerating *Brainbow* axolotl. The RFP[+] (marked with white arrows) and CFP[+] (pointed by orange arrows) mesenchymal chondrogenic cells give rise to transverse clonal chondrocytic columns. Note the remaining association of a CFP[+] perichondral cell and clonal chondrocytes during cartilage growth, as pointed by orange arrows. On the right, violin plots represent the predominant orientation of clones. The preferred direction of daughter cell allocation after cell division is longitudinally oriented in the mesenchymal chondrogenic cells (left violin plot) and transversally oriented in the maturing chondrocytes (right violin plot). Two-tailed *t*-test, ****p* < 0.0001. Each data point represents the orientation of a cell division, measured from *n* = 3 different limbs. Median and quartiles are represented as dashed and dotted lines, respectively. Lower panels illustrate how the diameter of regenerated cartilage increases with time, coinciding with the onset of transverse allocations of clonal progeny. The white line shows the border of the cartilage. The image at the bottom left of panel A illustrates the formation of clonal columns and corresponds to an area shown at high magnification from panel B at 18d.p.a.; cell repositioning is presented as individual planes from z-stack files from the same live-imaging series. **B** Cell dynamics in the regenerating digits (till 18d.p.a.) of genetically traced regenerating *Brainbow* axolotl. White arrowheads indicate an RFP[+] perichondral cell giving rise to a clone of chondrocytes. This area is magnified in the corresponding insets. Orange dotted line shows the area of chondrogenic mesenchymal cells at the tip of regenerating skeletal element. Green dashed line shows maturing chondrocytes in regenerating skeletal element.

regeneration in salamanders, the early regenerative steps in lizards also include activation of local periosteal progenitors that generate cartilaginous callus forming the proximal skeleton of the re-growing tail[39].

Following the further progression of skeletogenesis, we found that in contrast to development, skeletal regeneration is characterized by chondrogenesis and ossification, which are uncoupled in time compared to normal developmental growth. At the end of regenerative limb elongation, the freshly-shaped skeleton is composed of cartilage with only a minimal trace of bone, and the ossification advances only after the limb acquires near its final length. Furthermore, the bone starts to form from the outer surface of cartilage element, transiently representing cortical bone with cartilaginous core inside. This outside-to-inside ossification course resembles the one recently described in *Ambystoma* development[40]. However, it is highly dissimilar to the process of limb bone ossification in mammals, and rather resembles the ossification of mammalian Meckel´s cartilage[41].

Accordingly, ossification does not contribute substantially to the extension of the regenerating limb. These two steps (chondrogenesis and ossification) of skeletal regeneration differ from developmental skeletogenesis in salamanders, during which, according to our data, ossification starts long before the skeletal elements approach their final size, consistently with previous analyses[42–44]. Furthermore, our results revealed that regenerated skeletal segments are bulky compared to the contralateral counterparts which are shaped during development. This morphological distinction may reflect the observed uncoupling between cartilage and bone formation during regeneration, which may allow continuous expansion of cartilage in radial direction as supported by our mathematical modeling. From an evolutionary perspective, it is plausible to speculate that the bulkier and thicker skeleton might allow for a more stable appendage that can be weight-bearing during the regeneration process. In some cases, patterning anomalies are detected during regeneration in postmetamorphic *Pleurodeles* and *Ambystoma*, including altered shapes and numbers of skeletal elements in the wrist and fingers, which is in line with other reports[45,46]. The reasons for these imperfections are unknown and might be related to either external or internal environmental factors.

Orientation of chondrocyte cell divisions and controlled daughter cell repositioning play major roles in cartilage shaping and expansion[23], especially when scaling up a structure with complex geometry. Differences in proportions of differentially oriented cell divisions might account for the variation in the shape of regenerated versus non-regenerated skeletal parts. In the mammalian growth plate, the proliferative transiently amplifying cells arrange clonally along the main axis of bone growth[22] and fuel directional bone elongation. At the same time, in the developing vertebrate chondrocranium, multiple rounds of oriented cell divisions give rise to transversal clonal columns, providing for the thickness and lateral expansion of sheet-like cartilage through convergent extension[23]. We observed that the regenerating rod-shaped cartilages in the salamander limb expand via transversally oriented proliferation and clonal rearrangements, resembling basal sphenoid growth in mice. The live imaging of the developing phalange cartilage visualized this sequence of cellular events. Analysis of the clonal chondrocyte arrangements in other limb cartilages showed consistency with the cellular dynamics observed in phalange live imaging and with analyses of tissue-sections from regenerated ulna, radius and humerus.

In light of our results stemming from the proliferation and clonal arrangements, together with the mathematical modelling, the difference in the shape of skeletal elements generated during development vs regeneration might be explained by: 1) changing proportion and duration of longitudinal versus transversal cell divisions and 2) early onset of ossification within central parts of larval skeletal elements during development, which prevents further thickness expansion of cartilage in the ossifying regions. Therefore, since regenerating limb is clearly growing faster than developing limb, the regenerate manages to grow up before the ossification starts and constrains the central part of a skeletal element. Overall, our interpretation is that early embryonic ossification preserves the centers of larval cartilaginous rods in a thinner configuration, whereas the absence of early ossification in regeneration does not restrict the thickness increase in central parts of a skeletal rod. This results in bulky cylindrical bone structures in regenerates as compared to normally developed bones.

The clonal lineage tracing and proliferation analyses revealed de novo chondrogenesis at the distal ends of the skeletal elements, which fuels the massive outgrowth of the cartilaginous rods. Thus, the regenerative skeletal growth in salamanders depends on the continuous proliferation of mesenchymal progenitors with their subsequent differentiation towards chondrocytes. This mechanism of regenerating limb outgrowth resembles mammalian and avian limb bud growth, fueled by extensive proliferation of mesenchymal cells under the apical ectodermal ring[47]. However, the proliferative dynamics of chondrocytes differ from the one observed during growth of mammalian limb cartilage. The cellular mechanisms underlying elongation of the salamander rod-shaped cartilages appear similar to the cell dynamics found in mammalian basicranial synchondroses, where the expansion of the cranial base proceeds by cell repositioning due to convergent extension and hypertrophy before ossification occurs[23]. Basicranial synchondroses rely on slow proliferation rate of all chondrocytes, thus, depending on their embryonically deposited chondrogenic cells without extensive cell amplification cascades[23]. From this perspective, cartilage growth in salamander zeugopodial elements and mammalian synchondroses represent an evolutionarily more ancient mechanism of cartilage growth than the one characteristic of mammalian limbs, which rely on cartilage stem cells, their niche and extensive amplification cascade of the stem cell progeny[21,22]. Indeed, the stem cell niche and the cell amplification cascade in the mammalian limb growth plate might be considered recent evolutionary acquisitions and adaptations to the extensive limb growth required for large animal size and efficient terrestrial locomotion. In line with this view, the secondary ossification center, a structure facilitating formation of the stem cell niche[22], is absent in salamanders and appears to have evolved only in amniotes[48]. To sum up, elongation of both developing and regenerating salamander limbs involves

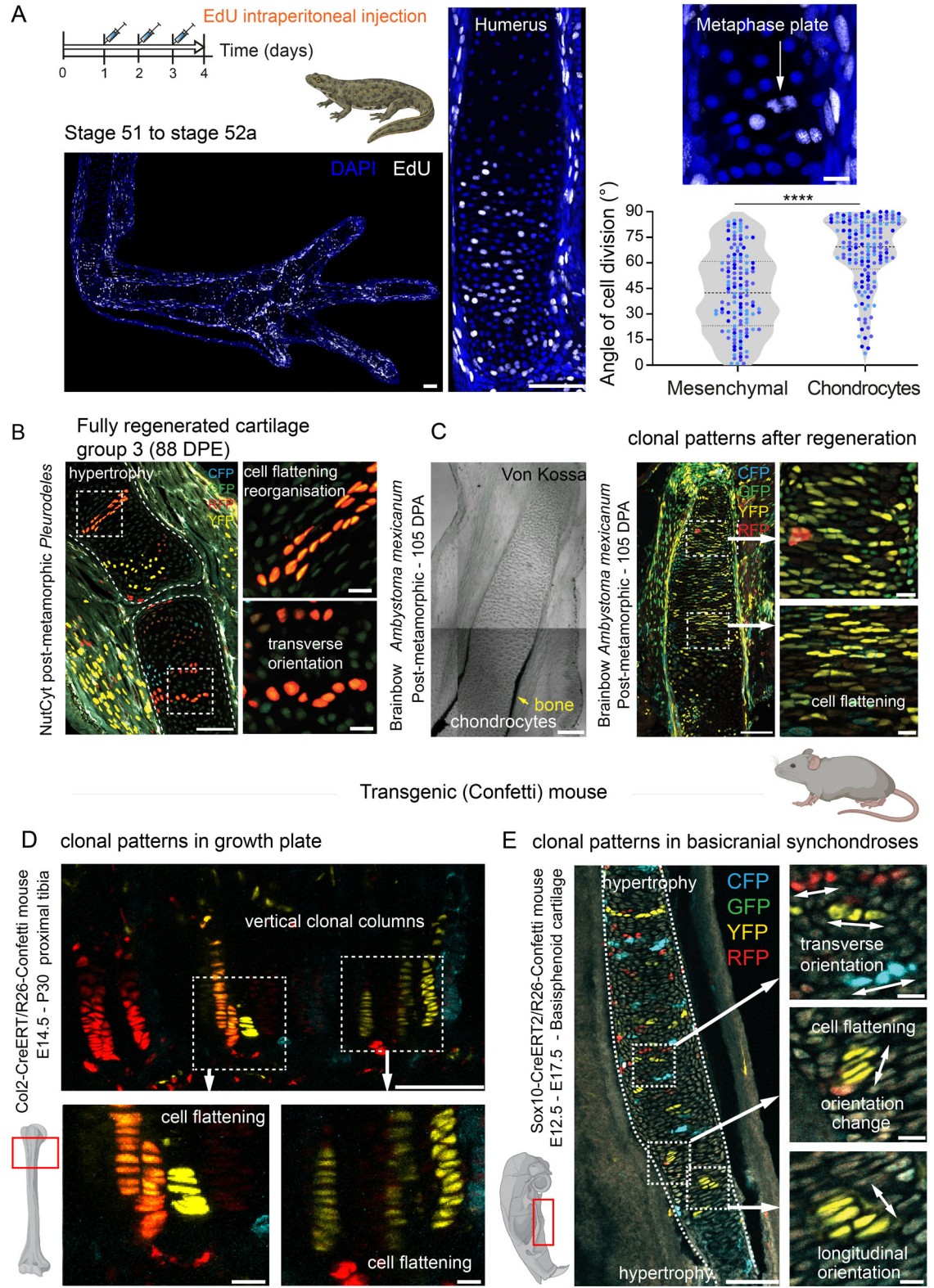

initially longitudinally-oriented proliferation of chondrogenic mesenchymal cells at the tip of the growing cartilage. The orientation of cell division is subsequently switched to transversal, which increases the diameter of the bone. This process does not involve the formation of a stem cell niche with cell amplification cascade and corresponding massive longitudinal extension like in mammals[22]. Instead, the skeletal growth in salamander limbs is achieved by mesenchymal progenitor proliferation and convergent extension-like repositioning of

chondrocytes during the flattening phase, similarly to mammalian basicranial synchondroses[23,49].

The *PTHrP/Ihh* loop is responsible for establishing the amplification cascade in the mouse growth plate and its longitudinal arrangement[24,50], whereas *Ihh* itself is coupling chondrocyte hypertrophy with bone ossification[32]. During regeneration of salamander skeletal elements, the *PTHrP/Ihh* loop[50] is established early during cartilage formation[4,51], but the spatiotemporal dynamics of expression

**Fig. 4 | Orientations of cell divisions and chondrogenic clones during salamander limb development and regeneration resemble the mammalian basicranial synchondroses. A** EdU pulse-chase performed at several time points (stage 51 to 52a is shown) in physiologically growing larval limbs of *Pleurodeles waltl* showed that the vast majority of chondrocyte cell divisions were oriented transversally. EdU-labelled doublets indicate cell division (shown in the right panel). Scale bar, 200 μm(left); 50 μm(center); 25 μm(right). The violin plots display differences in cell division/repositioning of cells found in rod-shaped skeletal elements of all species and processes analysed (see Fig. 2A, S5H, S6E). Two tailed *t*-test, ****p* < 0.0001. Each data point represents orientation of a cell division, measured from *n* = 3 different limbs; median and quartiles are represented as dashed and dotted lines, respectively. **B** Transverse orientation of clonal chondrocytes in regenerating long bones of post-metamorphic *Pleurodeles*. Cell flattening at ossification onset correlates with absence of longitudinally oriented clones typical for mammalian long bones (see Fig. S6A). **C** Von Kossa staining of bone tissue in the regenerating limb of an experimentally-induced post-metamorphic *Ambystoma mexicanum* at 105 d.p.a.

Yellow arrow points at mineralized layer with chondrocytes underneath. Next to it (right), neighboring section shows traced chondrocytes. **D** Proximal tibial growth plate is shown with the resting stem cell zone at the lower edge of the image. The recombination was induced at E14.5 in *Col2CreERT2/R26Confetti* embryos, when the limb is patterned, and the skeletal elements are made of stratified cartilage. The clones were analyzed at P30. A dotted line marks areas from magnified insets. Note the longitudinally-oriented chondrocytic clones containing proliferative flattened cells near hypertrophic zone. **E** Lineage tracing in mouse basicranial synchondroses highlights clonal arrangements. The recombination was induced at E12.5 in *Sox10-CreERT2/R26Confetti* embryos, and analyzed at E17.5. The basicranial cartilage undergoes ossification at E17.5 and allows observing cell dynamics in synchondroses. A dotted white line marks areas in magnified insets. Note the presence of transversally-oriented chondrocytic clones within basisphenoid, and cell flattening and repositioning near hypertrophic zone. The patterns in **D** and **E** were observed independently in 10 or more individual embryos from three litters. Scale bars are 100 μm, in small square magnified insets the bars are 10 μm.

patterns were not well defined. Our results showed that this signaling loop shows initially radial polarity, which subsequently transforms into longitudinal polarity along the growing axis. During the outgrowth phases of regeneration, the expression domains of *PTHrP*, *Gli1* and *Ihh* are enlarged compared to their contralateral controls. Such a process likely sustains regenerative limb growth, as the expression domains shrink to become like the contralateral control ones when the regenerate reaches the expected size. Thus, the molecular pattern of regenerating cartilage recapitulates the one observed during mammalian limb development[50,52], although the response of the target cells may be different. Interestingly, we also observed periskeletal (likely perichondrial) cells expressing *Gli1* and *PTHrP* in both regenerating and unamputated (normally growing) limbs. In mouse, such distribution is observed during early stages of embryonic limb development, i.e., embryonic days 14-16, whereas the *PTHrP*-positive domain becomes more restricted to the epiphyseal chondrocytes in postnatal mouse limbs. Of note, the early growth of mouse rod-like chondrogenic structures resembles the growth of basicranial synchondroses with the lack of amplification cascade and clonal intercalation and rearrangement[23]. Thus, this observation further supports the view that the growth of salamander long bones represents an ancestral mechanism of skeletal growth, where the *PTHrP/Ihh* system is already in place but acts differently as compared to the mammalian growth plate. Furthermore, the superficial ossification of Meckel cartilage, one of the most ancient skeletal structures in a vertebrate body, resembles the outside-in ossification of the salamander limbs[53].

In summary, our results establish an evolutionarily ancient nature of oriented clonal cell dynamics during cartilage growth and regeneration (Fig. 7). We report the absence of a chondrocyte amplification cascade in the developing, growing and regenerating salamander limbs. Furthermore, there is an uncoupling between cartilage growth and ossification in the regenerating limb, with the latter being much delayed. The balance between clonal orientation and expansion as well as delayed onset of cortical bone formation results in thicker bones with altered 3D geometry in the regenerated limbs in salamanders compared to normal development.

## Methods
### Animals and regulations
All animal experiments were performed according to European regulations and in consideration of Arrive Guidelines. Animal husbandry standardized methods were used for *Pleurodeles waltl*[64] and *Ambystoma mexicanum* (d/d)[55]. The clonal analyses performed in this study were performed with multicolour transgenic salamander and mouse lines reported previously: brainbow axolotl[16]; *Nucbow/Cytbow* Spanish ribbed newts[34], *R26Confetti* mouse coupled to *Sox10-CreERT2*[56] or *Col2-CreERT*[22]. *R26Confetti* (RRID:IMSR_JAX:017492) mice were received from the laboratory of Professor H. Clevers (Hubrecht Institute for Developmental Biology and Stem Cell Research), *Sox10-CreERT2* animals

were received from the laboratory of Professor Vassilis Pachnis (Francis Crick Institute), and *Col2-CreERT2* strain (RRID:IMSR_JAX:006774)[57] was received from the laboratory of S. Mackem, NIH). All animal (mouse) work has been approved and permitted by the Ethical Committee on Animal Experiments (Norra Djurförsöketiska Nämd, ethical permit N226/15 and N5/14) and conducted according to The Swedish Animal Agency's Provisions and Guidelines for Animal Experimentation recommendations. Experiments in *Pleurodeles waltl* were performed according to Swedish regulations. *Ambystoma mexicanum* experiments were done in accordance with the Saxony Animal Ethics Committee. A minimum of *n* = 2 *Pleurodeles waltl* per time point and a minimum of *n* = 3 *Ambystoma mexicanum* per time point and analysis were used. The Supplementary Data 2 contains information regarding the n number, sex and sizes for the different experiments.

### Limb amputations
*Pleurodeles waltl* were deeply anesthetized in tricaine methanesulfonate (MS-222, Sigma) diluted in chlorine-free water (0.02% for larvae, 0.1% for post-metamorphic animals) at pH=7.5. Developmental stages and animal sizes were documented. Proximal amputations were performed with a scalpel at the medial level of the humerus, and the protruding bone was trimmed with surgical scissors. The post-metamorphic animals were then transferred to individual containers with clean water containing sulfamethazine for recovery, where they were kept for 24 h. In larvae, sulfamethazine was not used. 24 h.p.a., the animals were kept in small groups at 25 °C and limb regeneration was surveilled daily. In order to compare results obtained in different developmental stages, both regenerating and contralateral control limbs were collected at the desired timepoints based on the external morphology of the regenerating limbs.

*Ambystoma mexicanum* were deeply anesthetized in a 0.007% benzocaine. Amputations to the limb were made with a sterile scalpel through mid-humerus, mid-radia/ulna, or distal to the final phalangeal joint in the case of digit amputations. Protruding limb bone was trimmed directly after amputation with spring scissors, and animals were allowed to recover and were monitored and assessed at regular time intervals after regeneration. A single injection of thyroxine experimentally induced metamorphosis in axolotls[55]. Juvenile axolotls (6 cm, snout to tail tip) and metamorphosed axolotls were amputated at the upper limb; the amputated limbs were collected immediately after amputation for analysis. The regenerated and the contralateral control limbs of juvenile and metamorphosed axolotls were collected approximately six months after the initial amputation.

### Live imaging during appendage limb regeneration
Double transgenic, *CAGGS::ER-Cre-ER-T2A-EGFPnls/CAGGS::Brainbow2.1*, larval axolotls measuring approximately 3.5 cm from snout to cloaca were recombined by bathing in (Z)−4-hydroxytamoxifen (Sigma) at

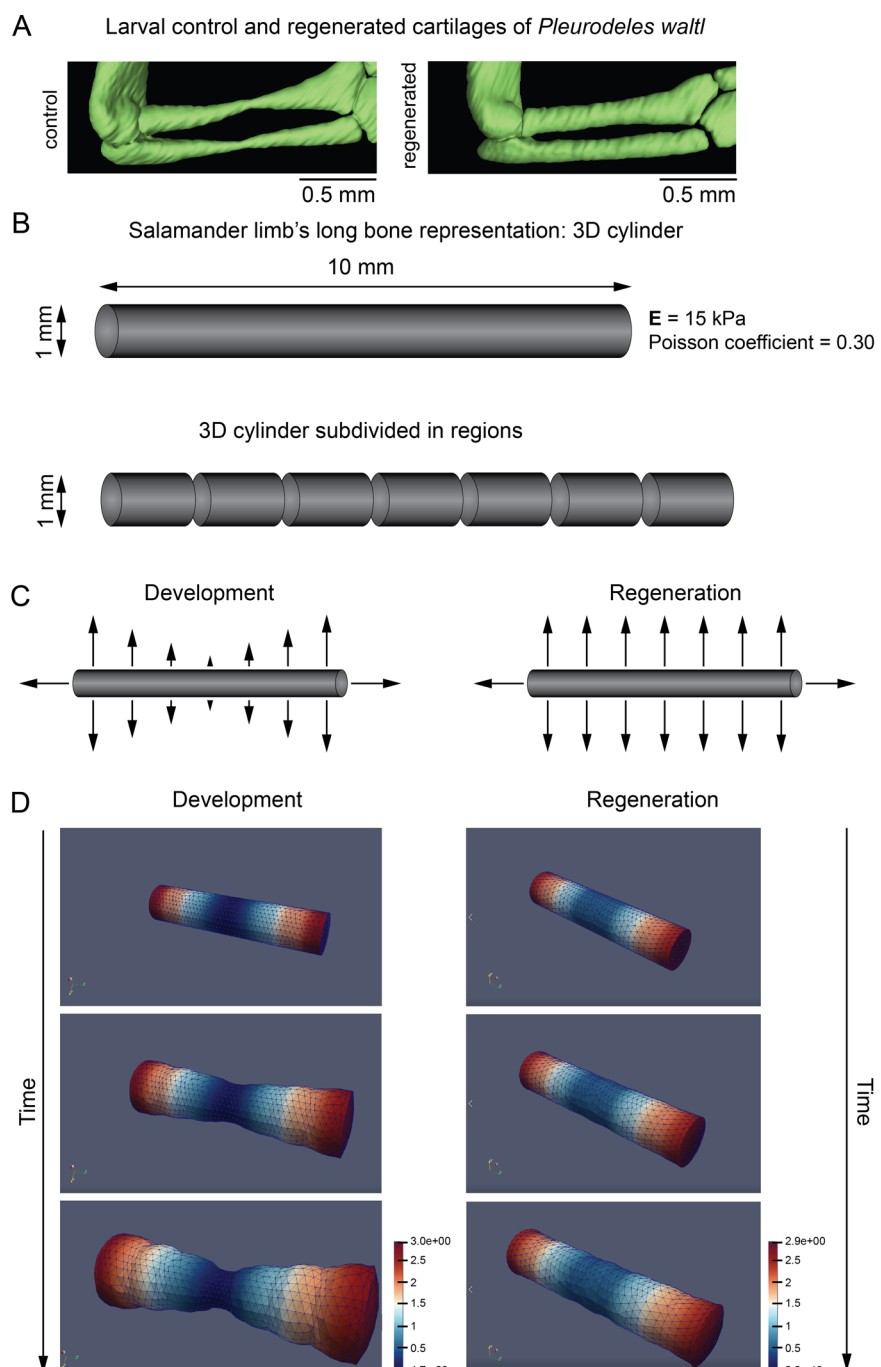

**Fig. 5 | Computational simulation of long bone shaping during development and regeneration. A** Differences in shape between normally developing (left) and regenerated (right) limb cartilages of the larval *Pleurodeles waltl*. **B** Long bone represented by a 3D cylinder. The geometrical and material properties used are specified. **C** Tissue growth is described as region-specific internal pressures. Tissues are subdivided into different regions, as shown in the scheme (**B**). Two configurations of internal pressures are applied for modelling normal growth (left) and regeneration (right). **D** Snapshots of the internal tissue growth simulation for the development (left) and regeneration (right) of the long bone shape.

concentrations ranging from 100 nM to 2 μM for durations of 30 min to overnight, for a range of cell labelling frequencies. Two weeks post tamoxifen treatment, animals were anesthetized with 0.007% benzocaine solution and placed in a glass-bottomed chamber (Willco) for imaging. Confocal stacks of approximately 80 μm depth were acquired using a Zeiss laser scanning confocal microscope every 24 h for 18-24 days following amputation of the digit. Technical details and information about transgenic animals can be found in[16]. The raw data from live imaging of salamander appendage regeneration has been previously used for different analyses than in this publication[8].

### Clonal labelling in transgenic *Pleurodeles waltl*
In order to label cells with a unique combination of fluorescent proteins, a self-excising Cre (seCre) plasmid from Loulier et al.[58] was electroporated in Nucbow-Cytbow transgenic *Pleurodeles waltl* ($Pw^{NucCyt}$;[34]). The seCre plasmid was purified using Qiagen Maxiprep and resuspended in DNAse-free distilled water. The electroporations were performed with a NEPA21 electroporator (Nepagene) using tweezers with round platinum plate electrodes (Nepagene, CUY651P for post-metamorphic animals and CUY615 for larvae). First, bilateral amputations were performed at stages 54 (*n* = 5), 55a (*n* = 6), 56 (*n* = 5)

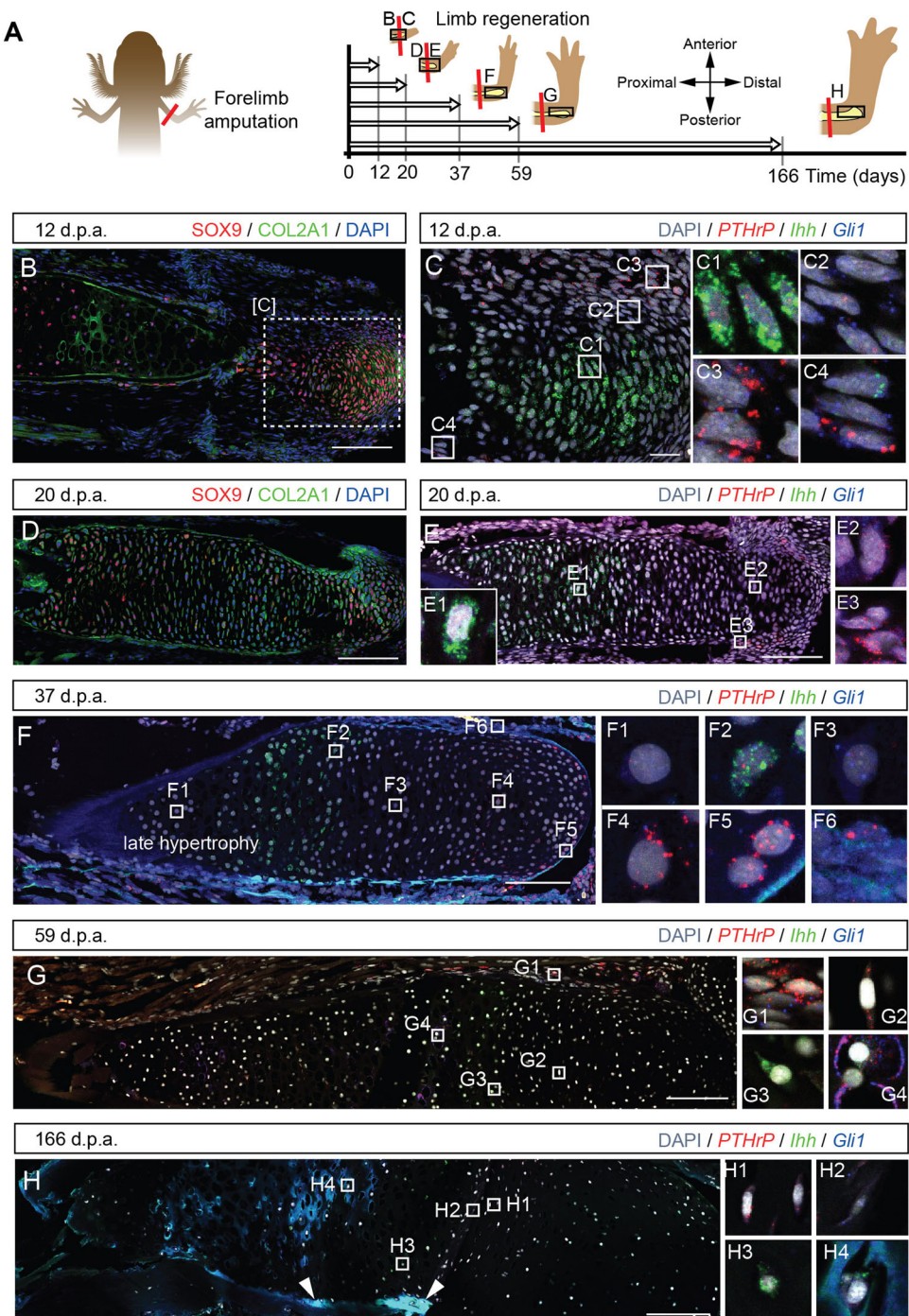

**Fig. 6 | Dynamic expression of the PTHrP-Ihh loop components during skeletal regeneration. A** Experimental outline for assessment of the *PTHrP-Ihh* loop during regeneration in *Pleurodeles waltl*. Larvae at stage 54 underwent unilateral amputations. The regenerating limbs and contralateral controls were collected at the selected time points. Note that the axis drawn in A applies to all the pictures of the regenerating humerus in the figure (**B–H**). **B** At 12 days post-amputation (d.p.a.), the core of the blastema showed SOX9⁺/Col2A1⁺ emerging chondrocytes. **C** At 12 d.p.a., a central group of *Ihh*⁺ (C1) is wrapped by a layer of *Gli1*⁺ (C2) cells followed by a layer of *PTHrP*⁺ cells (C3). This pattern was also observed in periskeletal cells surrounding the stump bone (C4). **D** At 20 d.p.a., the expanding humerus wrapped the stump bone and consisted of SOX9⁺/Col2A1⁺ chondrocytes. **E** At 20 d.p.a., *Ihh*⁺ prehypertrophic chondrocytes occupy a wide region proximal to the amputation plane (E1), while double-labelled *Gli1*⁺/*PTHrP*⁺ cells were found both in the distal portion of the humerus (E2) and in the perichondrium (E3). **F** At 37 d.p.a., we detected the first chondrocytes devoid of *PTHrP-Gli1-Ihh* (F1). *Ihh*⁺ expression was

maintained in the pre-hypertrophic chondrocytes (F2), followed by *Gli1*⁺ cells (F3). *PTHrP*⁺ was strongly detected in periarticular chondrocytes in the epiphysis (F4, F5). Perichondrial cells expressed *PTHrP* and *Gli1* at this stage (F6). **G** At 59 d.p.a., *PTHrP* and *Gli1* perichondrial cells were still present (G1). The *PTHrP*⁺ expression was reduced in the epiphysis (G2), and the *Ihh*⁺ expression domain occupied most of the humerus (G3). We observed the first hypertrophic chondrocytes in the regenerates containing *PTHrP* and *Gli1* puncta (G4). **H** At 166 d.p.a., the expression patterns of *Ihh*, *PTHrP* and *Gli1* in the cartilage resembled those found in the contralateral control limbs. The *PTHrP*⁺ expression was restricted to fewer cells (H1), followed by scarce *Gli1*⁺ cells (H2) and further the *Ihh*⁺ cells (H3). We observed multiple hypertrophic chondrocytes in patches of the regenerate containing *PTHrP* and *Gli1* puncta (H4). Arrowheads point to ossification occurring in patches. Scale bars: 200 μm (**B**, **D–H**) and 50 μm (**C**).

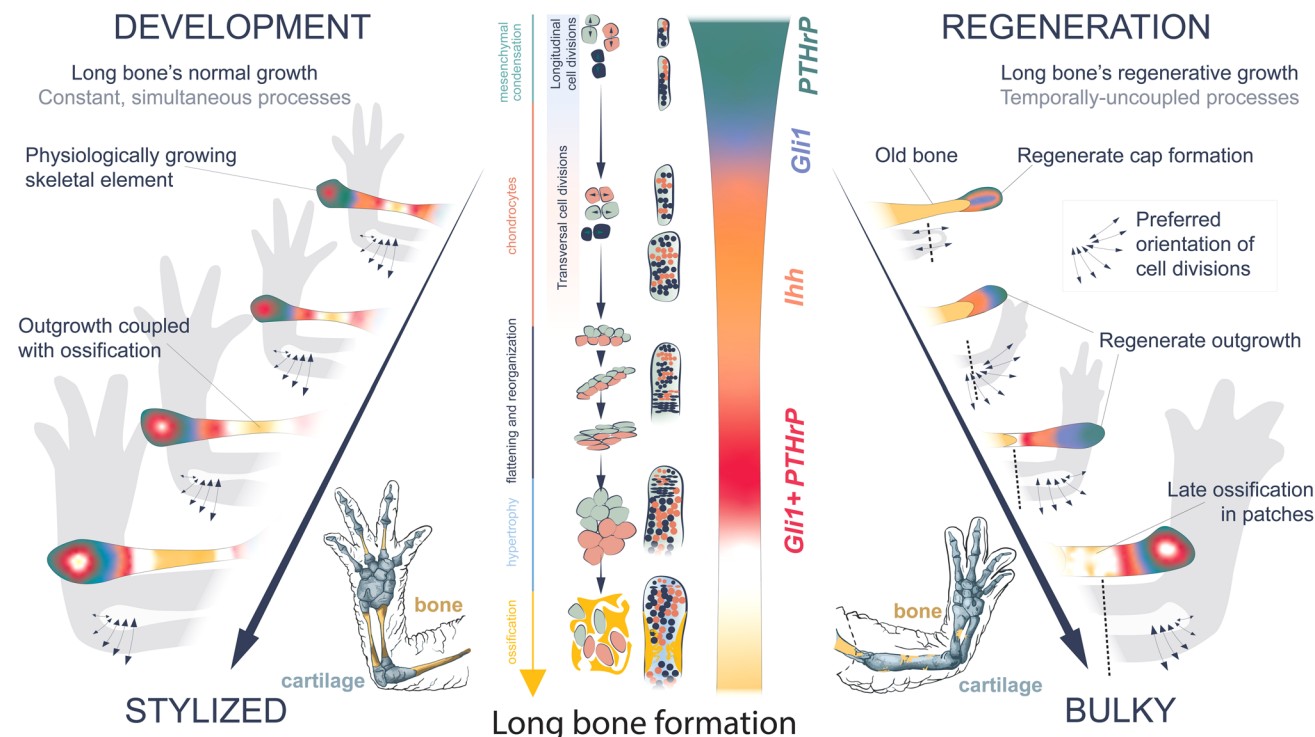

**Fig. 7 | Graphical Abstract.** Summary of similarities and differences between skeletal development and regeneration.

(Fig. S6) and 54 ($n = 13$) (Fig. S7). When the blastema acquired the cone-shape stage (7 d.p.a. for larvae and 10 d.p.a. for post-metamorphic animals), the anesthetized animals were placed in ice-cold PBS, and each limb was held with plastic-coated forceps for plasmid injection into the blastema followed by bi-directional (+/-) electroporation. For larvae, the machine was set to give two poring pulses at 35 mV/cm of 50 ms, with an interval of 10 ms between pulses, followed by 6 transfer pulses at 25 mV/cm of 50 ms, with an interval of 50 ms between pulses. For post-metamorphic juveniles, the machine was set to give two poring pulses at 70 mV/cm of 50 ms, with an interval of 25 ms between pulses, followed by 6 transfer pulses at 50 mV/cm of 50 ms, with an interval of 15 ms between pulses. Labelled cells in the blastemas were assessed three days after electroporation. Animals with labeled clones were kept until full regeneration when the tissue was collected for clonal analysis ($n = 4$-5 animals per group).

### Tissue processing

After limb amputation, the samples were placed into freshly prepared 4% paraformaldehyde (PFA) and fixed for 24 h (or 48 h for RNAscope) at +4 °C on a roller. Samples were subsequently cryopreserved in 30% sucrose (VWR C27480) overnight at +4 °C, embedded in either gelatin[34] or OCT media (HistoLab 45830) and sections cut between 10 μm to 40 μm on a cryostat (Microm). Mouse tissue was processed identically; all mouse tissue samples were embedded in O.C.T. only. If needed, sections were stored at −20 °C (or −80 °C for RNAscope) after drying or processed immediately after sectioning. For imaging, washed (15 min in PBS) slides with sections were mounted with 87% glycerol mounting media (Merck).

### EdU incorporation analysis

For the experiments in *Pleurodeles*, EdU (Life Technologies) was injected by intraperitoneal injections at selected stages (dose: 50 μg per gram of body weight). For the study of limb development (Fig. 4A), 3 pulses of EdU were administered every 24 h in larvae between stages 51 and 53 ($n = 3$ animals). 24 h after the last injection, both forelimbs were fixed and processed for EdU detection. For the EdU chase

experiment during regeneration (Fig. S5B–D), stage 52-53 animals ($n = 4$) were amputated unilaterally, keeping the contralateral limb as a control for development. Starting at 13 d.p.a., the animals received a single pulse of EdU every third day (total: 5 pulses), and both limbs were collected and fixed for analysis two days after the last pulse (27 d.p.a.). EdU detection was performed according to[34].

For the experiments in *Ambystoma* (Fig. S5A), the animals were anesthetized in a 0.007% benzocaine solution and given a single intraperitoneal injection of 5-Ethynyl-2′-deoxyuridine (EdU) equal to 10 μg per gram of body weight from a stock EdU concentration of 0.5 mg/ml. Two days post-injection, limbs were harvested and fixed in 4% paraformaldehyde before being processed for EdU visualization and downstream imaging.

### Microscopy, image analysis and quantifications

Confocal microscopy was performed using LSM700, LSM780 CLSM and Zeiss LSM880 Airyscan CLSM instruments. The settings for imaging Brainbow fluorescent proteins were identical to those previously described in Confetti mice[56]. The imaging of the confocal stack was done with a Zeiss LSM780 CLSM, Plan-Apochromat 3 10x/0.45 M27 Zeiss air objective. Axolotl digit imaging was performed on a Zeiss LSM780 inverted confocal microscope using a 20x/0.8NA Plan-Apochromat objective. Fluorophores were acquired as sequential image volumes. Image stacks were manually merged and aligned post-imaging using Fiji image analysis software[59]. Axolotl digit regeneration imaging was supported by the Light Microscopy Facility, a core facility of BIOTEC/CRTD at Technische Universität Dresden.

### Cell orientation measurement and graphic representation

Angles of cell divisions were measured manually using Screen Protractor (Iconico, Philadelphia, USA) and ImageJ, and violin plots were created using GraphPad Prism 9.1.0.

### Histological staining

Slides were stained for mineral deposition using von Kossa calcium staining: 5% silver nitrate solution was added to the sections at room

temperature and exposed to intense light for 30 min. After that, the silver nitrate solution was removed, and the slides were washed with distilled water 3 times for 2 min. 2.5% sodium thiosulphate solution (w/ v) was added to the sections and incubated for 5 mins. Slides were again rinsed 3 times for 2 min in distilled water. The sections were then counterstained using Alcian blue. Alcian blue solution (0.1% Alcian blue 8GX (w/v) in 0.1 M HCl) was added to the tissue for 3 min at room temperature and then rinsed 3 times for 2 min in distilled water. Slides were then dehydrated using an ethanol gradient (70%, 95%, and twice in 99.5%, for 5 min each. Finally, the slides were incubated in two xylene baths (for 2 min and then for 5 min) before mounting and analysis.

### Tissue contrasting for micro-CT scanning
Our staining protocol has been modified from the original protocol developed by Brian Metscher's laboratory (University of Vienna, Austria)[27]. After dissection, the limbs were fixed with 4% aqueous solution of formaldehyde in PBS for 24 h at +4 °C, with slow rotation. Samples were then dehydrated by incubation in incrementally increasing concentrations of ethanol in PBS (30%, 50%, 70%); samples were incubated at +4 °C for 1 day in each concentration to minimize the tissue shrinkage.

The samples were contrasted with 1.0% PTA (Phosphotungstic acid, Sigma Aldrich) in 90% methanol. After sample dehydration, the tissue-contrasting PTA solution was added to the samples and changed daily with the fresh solution. Depending on the size of the sample, the 1% PTA contrasting mixture was applied for 1 week (primarily larval limbs, up to 1 cm of length) or up to 3 weeks (adult long-term regenerated limbs, robust and over 1 cm) to ensure a sufficient penetration of the contrasting agent. Subsequently, the contrasted limbs were rehydrated through a methanol gradient (90%, 80%, 70%, 50% and 30%) in sterile distilled water. Rehydrated limbs were embedded in 1% agarose gel (A5304, Sigma-Aldrich) and placed in polypropylene conical tubes to avoid motion artefacts during X-ray computed tomography scanning.

### Micro-CT analysis (micro-computed tomography analysis)
The micro-CT analysis of the limbs was performed using the laboratory system GE phoenix v|tome|x L 240 (GE Sensing & Inspection Technologies GmbH, Germany), equipped with a 180 kV/15 W maximum power nanofocus X-ray tube and high contrast flat panel detector DXR250 with 2048 × 2048 pixel, 200 × 200 μm pixel size. The exposure time was 900 ms in all positions over 360°. Three projections were averaged in each position to reduce the noise in the data. The micro-CT scan was carried out at 60 kV acceleration voltage and with 200 μA X-ray tube current. The voxel size of obtained volumes appeared in the range of 1.5 μm–13.0 μm, depending on the size of the limb. The tomographic reconstructions were performed using GE phoenix datos|x 2.0 3D computed tomography software.

The cartilage in the limb was segmented manually using Avizo (Thermo Fisher Scientific, USA) – 3D image data processing software. Manually segmented models were then transferred to STL models and imported to VG Studio MAX 3.4 (Volume Graphics GmbH, Germany) for further visualization[60]. Shape comparison of limbs was done in VG Studio MAX 3.4 by Nominal/actual comparison module.

### Immunohistochemistry
Immunohistochemistry was performed according to[34]. The primary antibodies mouse anti-COL2A1 (1:250, DSHB, II-II6B3) and rabbit anti-Sox9 (1:250, Cell Signaling, D8G8H/82630 S) were used. COL2A1 antibody from hybridoma bank has been validated in multiple species including amphibians. We predicted its specific binding to *Pleurodeles waltl* COL2A1 protein in silico based on homology of the antibody epitope [chicken COL2A1 triple helix domain (aminoacids 491-586)] with the *Pleurodeles* homologue region (88% identity). The staining obtained in our results in salamanders was consistently found to be specific to cartilage elements. Rabbit anti-SOX9 antibody from Cell Signaling is predicted to cross react with mouse, rat and human samples. The specificity in *Pleurodeles waltl* was first predicted in silico based on homology of the antibody epitope [amino terminus of human Sox9 (aminoacids 1-188)] with the *Pleurodeles* SOX9 sequence (93% identity). We then validated the staining empirically by the specific signal found in the cartilage elements as well as the ependymoglial cells in the CNS.

### Single-molecule RNA in situ hybridisation
RNA in situ hybridisation experiments were performed using RNAscope®, an RNA in situ hybridisation technique described previously[61]. Paired double-Z oligonucleotide probes were designed against target RNA using custom software (see Supplementary Data 1 for details on the sequences used for probe design). The following probes were used: Pwa-GLI1-C3, cat. no. 823081-C3; Pwa-IHH-O1-C2, cat. no. 823041-C2; Pwa-PTH-C3, cat. no. 823061-C3; Pwa-PTHrP, cat. no. 823071; Pwa-SHH-O1, cat. no. 823051. The RNAscope® Multiplex Reagent Kit (Advanced Cell Diagnostics (ACD), Newark, CA, cat. no. 323110) with hydrogen peroxide/protease reagents (cat. no. 322381, ACD) and wash buffers (cat. no. 320058, ACD) was used in combination with TSA Plus Cy3 (cat. no. NEL744001KT, Perkin Elmer) and TSA Plus Cy5/fluorescein (cat. no. NEL754001KT, Perkin Elmer). Negative control background staining was evaluated using a probe specific to the bacterial dapB gene. Counterstaining was performed with DAPI in PBS. Fluorescent images were acquired using an LSM700 confocal microscope using a 20x objective.

### Simulation of limb growth using the finite element method
The growth of a salamander limb during development or regeneration is a biological process driven by cell division. These dividing cells generate internal forces pushing other cells and elongating the tissue. Limb growth is a slow process that takes a few months to complete. Therefore, it is possible to approximate the limb with an elastic and homogeneous material that has a cylindrical shape. To study limb growth during development and regeneration, we consider a 3D cylinder and subdivided it into different regions. We applied a force field to each region to model the effects of cell divisions. Each of these forces consists of axial and radial components. We considered all regions to be subject to force fields that have the same modulus during regeneration. While during development, we gradually increased the moduli of radial components as we moved from the regions in the center to those closer to the sides. As a result, we obtained a linear elasticity problem that can be solved using the finite element method. To achieve this, we began by generating a tetrahedral mesh corresponding to the computational domain. We generated this mesh using the NETGEN algorithm in the Salome-Meca software[62]. After that, we used the open-source package Code-Aster[63] to perform finite element simulations. After loading the generated mesh, we assigned the mechanical forces and material properties. The CPU time for each simulation with this model was 3.31 s. The post-processing and visualization of results were performed using the software ParaView.

### Figure preparation
3D-reconstructions and CT segmented images were exported from Avizo/VG Studio MAX 3.4. Raw microscope images were adjusted for brightness and contrast using ImageJ. The graphs were prepared using GraphPad Prism 9.1.0. All the schematic drawings were done by the authors using Adobe Illustrator. The final figure panels were prepared using Adobe Illustrator.

### Statistics & reproducibility
Due to the exploratory nature of this study, no statistical methods were used to predetermine sample size. This study includes analysis of multiple developmental and regenerative stages of two salamander

species. The similar observations obtained through independent experiments performed in developing, regenerating, larval and post-metamorphic salamanders of the two species support the sufficiency and reproducibility of our experiments (sample sizes at least $n = 3$ limbs per experiment). One limb was excluded from the analysis due to bad quality staining. Detailed information on the experimental animals used and the related figures are provided in the Supplementary Data 2 to facilitate reproducibility. Both male and female salamanders were used in all experimental conditions. For experiments done in larvae, where sex could not be determined, animals are reported as unknown. Clonal patterns in mouse were observed in at least 10 individual embryos or pups from three independent litters, sex was not determined. The investigators were not blinded to allocation during experiments and outcome assessment.

### Reporting summary

Further information on research design is available in the Nature Research Reporting Summary linked to this article.

## Data availability

A computational mesh was generated for the 3D cylinder using the NETGEN algorithm in the open-source software Salome-Meca[62]. The mesh consists of 65930 tetrahedral elements. Numerical simulations for the linear elasticity problem were conducted using the finite element method package Code-Aster[63]. The software ParaView was used to post-process the obtained results. All the information necessary to reproduce the simulations is provided in the Simulation of limb growth using the finite element method in the Methods Section. All other relevant data supporting the key findings of this study are available within the article and its Supplementary Information files or from the corresponding author upon reasonable request. The quantitative data (measurements of cell orientation, bone volume) generated in this study are provided in the Source Data file. Source data are provided with this paper.

## Code availability

The open-source package Code-Aster was used in this study (Delbecq, J. M. The Aster Code, 1999).

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

## Acknowledgements

I.A. was supported by ERC Consolidator grant STEMMING-FROM-NERVE and ERC Synergy grant KILL-OR-DIFFERENTIATE 856529, Knut and Alice Wallenberg Foundation, Swedish Research Council, Bertil Hallsten Foundation, Cancerfonden, Paradifference Foundation. A.S. was supported by the Swedish Research Council, Cancerfonden, Olle Engkvists Stiftelse. M.K. was supported by the Max Planck Society. A.E. was supported by NIH Ruth Kirschstein postdoctoral fellowship F32GM117806. E.M.T. was supported by the ERC AdG program and DFG FZT111. M.T., T.Z. and J.K. acknowledge the project CEITEC 2020 (LQ1601) with financial support from the Ministry of Education, Youth and Sports of the Czech Republic under the National Sustainability Programme II and CzechNanoLab Research Infrastructure supported by MEYS CR (LM2018110). M.T. acknowledges the Brno City Municipality as a Brno Ph.D. Talent Scholarship Holder and Martina Roeselova Memorial Fellowship. J.F.F. was supported by the National Key R&D Program of China 2019YFE0106700, the Natural Science Foundation of China 31970782. P.T.N. was supported by the Swedish research council (#2019-01919) and Karolinska Institute. A.S.C was supported by The Swedish Research Council (#2020-02298) and The Russian Foundation for Basic Research (#19-29-04115). We thank Olga Kharchenko for the help with illustrations. Visuals in Fig. 4 were created using BioRender.com.

## Author contributions

Conceptualisation: M.Kau., A.J.A., A.S. and I.A. Methodology: M.Kau., A.J.A., M.T., Z.Y., P.N., J.P., A.H. and J.C. Investigation: M.Kau., A.J.A., M.T., Z.Y., P.N., J.F., A.E., M.Kav., J.P., T.Z., A.H., J.B. and J.C. Figure preparation: M.Kau., A.J.A., M.T. Writing – Original Draft, M.Kau. and A.J.A. Writing – Review & editing: M.Kau., A.J.A., A.S.C., A.S. and I.A. Funding acquisition: E.T., J.K., A.S. and I.A. Resources, supervision: A.B.

## Funding

## Competing interests

The authors declare no competing interests.
