## [Peer Review File · Nature Communications]

REVIEWER COMMENTS

Reviewer #1 (Remarks to the Author):

This MS compares the clearly delineated cell and molecular events that occur during limb development to the process of salamander limb regeneration. The authors convincingly demonstrate that these 2 processes are in fact dissimilar; limb skeletogenesis is driven by synchronous cartilage to bone transitions regulated by the growth plate, while limb skeleton regeneration occurs by the temporal separation of chondrogenesis followed by osteogenesis. An additional difference found by the authors is the lack of cartilage stem cells in the regenerate. These findings are interesting and novel and provide a starting point for others to further investigate the signaling pathways mediating these divergent processes. It is somewhat surprising that the authors, once establishing that the processes are different, did not look for changes in signaling molecules that might direct the cell behaviors described. As it does not appear that the PTHrP/Ihh/Gli1 axis can explain the temporal separation of chondrogenesis and osteogenesis, the authors might consider other pathways that are known to play significant roles in directing cartilage and bone formation including FGFs, Wnts and BMPs. There might also be novel signals involved in the process that could be identified by profiling the tissue regenerates. This type of additional information would increase the significance to the current MS by adding more mechanistic data.

--

Reviewer #2 (Remarks to the Author):

It is generally understood that the process of organ regeneration is a simple repetition of developmental process. It is highly appreciated that the authors questioned this point and conducted research from a new perspective.

In addition to comparing mammal (mouse) and amphibia, the authors are trying to generalize the phenomenon, by using 2 salamander species. In fact, the comparison data of development and regeneration is very interesting.

This research, of course, seeks to elucidate the mechanism of regeneration. But at the same time, it is also valuable paper that provides basic information on limb regeneration of salamander.

However, I require some responses to these questions and comments, before acceptance to Nature Communications.

Major comments

1. Each of the two studies (ossification patterns of the bones, and clonal analysis in finger cartilage) is valuable. However, the relationship between each study is unclear. Can the phalange formation mechanism explain the regenerative capacity of the entire arm, including the humerus?

2. The micro-CT is a new technology for regeneration research in the salamanders. The authors need to show how accurately distinguish the areas of the cartilage and the bones by analyzing the micro-CT data

3. Although the sample number of regenerative limbs is small (maybe 3?), are the observed phenomena common for regenerative process? Can you quantify the area of ossification?

4. EdU is not sufficient for the lineage trace (Fig. 6). Is it difficult to combine NucCyt and EdU labeling? Alternatively, show the M-phase figure of Edu-positive cells. If the direction of cell division is known from the direction of the mitotic spindle, it will help the reader's understanding.

Minor comments

1. Maybe, it is a typo. Fig. 1A, middle bottom "Adult stage 56".

2. In Fig. 2A, "dpa" is displayed above the un-regenerating control. It is a little bit confusing as these limbs did not be amputated.

3. Also, in Fig. 2A, only "regenerating limb 10 dpa" has been expanded. It is still confusing, despite the authors show it with a scale bar.

4. How do you distinguish the plane of amputation? I guess, it is shown as the bone misalignment like regenerating limbs in Fig. 2A. If so, it looks strange in the regenerating limb (st 55) in Fig. 2B right. Also, correct the red-dotted line extending to the palm.

5. Fig.4 and Fig. 5 are look very busy. I recommend organizing the layout of the panels and how to add captions. Especially, the direction of the rose diagram is difficult to read.

Best Regards,

--

Reviewer #3 (Remarks to the Author):

The paper analyzes the way bones are formed during salamander limb regeneration. By μ CT the authors find that -in contrast to mammalian limb development - the regenerating skeleton is formed by chondrocytes, which are only replaced by bone after the final size of the regenerating limb is reached. Using genetic, multicolor fate mapping tools (brainbow) they show that after condensation chondrocytes align horizontal to the long axis of the skeletal element mimicking basicranial synchondrosis development more than that of mammalian long bones. They further investigate Sox9/Col2 and Ihh/PthrP expression, which are expressed in the expected regions of the cartilage anlagen

The differences to the embryonic and postnatal differentiation of mammalian long bones are very interesting observations and of importance towards an understanding of skeleton regeneration. However, while the paper if nicely describes and the analyses bone formation by μ CT and cell division planes in genetically labeled cells the experiments are quite redundant. While the investigation of two species at pre- and post-metamorphic stages each, clearly supports the findings, it makes reading quite repetitive. Similarly, the figures are quite repetitive analyzing ossification by μ CT: fig. 1-3 & fig. S1, S4 division plane fig. 4-6 and fig. S3). It might make more sense to restrict the analysis in the paper to one species and include the other data in the supplements.

In fig. 4b and 5c the authors investigate developing bones. These seem to undergo transverse division planes similar to the regenerates indicating that the plane of division is characteristic for Salamanders rather than regeneration. Some information about the stages analyzed and how they relate to mouse development would also be important. Fig. 5 CD show sections through synchondroses and long bones of mice. Why are the stages analyzed different (pre and postnatal) and how do they relate to the investigated Salamander stages?

Fig. 8; fig. S5: The analysis of Sox9/Col2/Ihh/Gli/PthrP expression is interesting, but it is not clear what the authors want to conclude? That the pathways act in the same way? How does that contribute to or explain the transverse division planes?

The expression is difficult to see especially at later stage. It would be helpful to display the original (not combined) pictures at least as supplement, to better evaluate the expression of single genes.

Are Sox9 and Col2 exclusively expressed (fig. 8; fig. S2, S5)? This seems to be surprising.

Reviewer #4 (Remarks to the Author):

The presented article shows comparative data on the ossification process in the context of limb development vs. limb regeneration in salamander models. The findings here demonstrate marked differences between limb development and limb regeneration that help us to understand one of the long-standing questions as to whether regeneration is a recapitulation of the developmental program and whether the cellular processes involved in the morphogenesis of regenerated tissues mirror the events of embryonic development. On the other hand, the findings presented here show interestingly the conservation of growth and ossification mechanisms between regenerating limbs and the development of craniofacial structures such as synchondroses at the cranial base.

RESPONSE TO REVIEWER COMMENTS

Reviewer #1 (Remarks to the Author):

This MS compares the clearly delineated cell and molecular events that occur during limb development to the process of salamander limb regeneration. The authors convincingly demonstrate that these 2 processes are in fact dissimilar; limb skeletogenesis is driven by synchronous cartilage to bone transitions regulated by the growth plate, while limb skeleton regeneration occurs by the temporal separation of chondrogenesis followed by osteogenesis. An additional difference found by the authors is the lack of cartilage stem cells in the regenerate. These findings are interesting and novel and provide a starting point for others to further investigate the signaling pathways mediating these divergent processes. It is somewhat surprising that the authors, once establishing that the processes are different, did not look for changes in signaling molecules that might direct the cell behaviors described. As it does not appear that the PTHrP/Ihh/Gli1 axis can explain the temporal separation of chondrogenesis and osteogenesis, the authors might consider other pathways that are known to play significant roles in directing cartilage and bone formation including FGFs, Wnts and BMPs. There might also be novel signals involved in the process that could be identified by profiling the tissue regenerates. This type of additional information would increase the significance to the current MS by adding more mechanistic data.

- We would like to thank the reviewer for the valuable suggestions. We completely agree that it is very important to understand the molecular signals that switch on the timing of osteogenesis or are responsible for differences in chondrocyte cell division orientation. We appreciated the importance of this task and took on a challenge by screening regenerating salamanders with inhibitors of major signalling pathways during this revision. After the regeneration, we contrasted 75 control and experimental limbs with PTA and subjected all of them to microCT and 3D reconstructions. That was a *tour de force*, which depleted the colony of salamanders and took a very long time to accomplish (we hope the reviewer will appreciate the scale of this effort). Although we could influence the length of the regenerating limbs, we did not find a signal responsible for the uncoupling of chondrogenesis and osteogenesis (changing their resulting proportions and timing of ossifications), or for orienting cell divisions in emerging cartilage. The newly generated results are presented as plots in Figure S9. The failure of this approach can be explained by the potential combinatorial action of multiple ligands, which is beyond our financial capacities to address and definitely beyond the scale of this revision, as it might take years of work and a large number of salamanders.

Furthermore, after an extensive literature screen and published single-cell data re-analysis, we have selected several essential signalling molecules that could be driving the control of cell division orientations and osteogenesis, being also differentially expressed in development and regeneration. For this, we performed new RNAscope experiments on developing and regenerating salamander limbs. However, we did not find any additional pattern that would further clarify the differences in development vs regeneration (data not included in the manuscript, however, we present examples here – please, see the images after the end of the response letter).

After finishing the experiments mentioned above, we additionally focused on better explaining the PTHrP-Ihh loop and corresponding difference in developing and regenerating limbs, which is now also shown in a new concluding Figure 7 and Figure S3, Figure S8, and described in Results:

“... we showed that PTHrP/Ihh loop has a dynamic expression pattern during regeneration, in contrast to the gradient maintained during developmental growth in salamanders. The continuous endochondral ossification during development shows a constant pattern of moderate expression of PTHrP in the proliferative zone, intermediate Gli1, and Ihh expressed in the chondrocytic cells (Fig. S3C). In contrast, our analyses revealed that PTHrP/Ihh loop has a dynamic expression domains during regeneration that vary in size and shape compared to their contralateral control limbs (Fig. 6, Fig. S8). First, cartilage condensation is characterised by a switch to a radial polarity pattern of Ihh-Gli1-PTHrP from the core to the periphery (Fig. 6B, C). Second, Ihh+ pre-hypertrophic chondrocyte domain expands and forms the callus wrapping the amputated bone, while the growth continues distally, characterised by Gli1+ columnar chondrocytes and PTHrP+ articular chondrocytes (Fig. 6D, E). Third, the gradient observed in development is restored but with extended expression domains compared to the unamputated controls (Fig. 6F). Finally, we found Gli1 and PTHrP double-positive signal in enlarged chondrocyte cells located proximally to the flat cells (Fig. 6G4), which are likely hypertrophic chondrocytes (both in developing and regenerating limbs). We also found periskeletal cells positive for Gli1 and PTHrP, which could be contributing to the developing as well as the regenerated cartilage (Fig. 6C⁴, E3, F6, G1, S8B⁴).”

We also revised the overall data presentation and expanded on the experiments requested by other reviewers, providing more data on microCT of regenerating limbs (Fig. 1), adding counts of oriented cell divisions in normally developing limbs (Fig. 4, Fig. S5), performing double EdU and clonal tracing tracking of chondrocytes in regenerating cartilage (Fig. S7), and many more.

Although we did not manage to identify the complex combinatorial effect of the molecular mechanisms responsible for the difference in development and regeneration, we hope the reviewer will appreciate the scale of our efforts to understand this phenomenon, and will agree with us that any further research would be a task for a whole new project and several years of intense work.

When it comes to the analysis of cell dynamics and skeletal shaping in regeneration and development, we sincerely hope that by adding additional experimental data we strengthened the major observations and conclusions.

Reviewer #2 (Remarks to the Author):

It is generally understood that the process of organ regeneration is a simple repetition of developmental process. It is highly appreciated that the authors questioned this point and conducted research from a new perspective. In addition to comparing mammal (mouse) and amphibia, the authors are trying to generalise the phenomenon, by using 2 salamander species. In fact, the comparison data of development and regeneration is very interesting. This research, of course, seeks to elucidate the mechanism of regeneration. But at the same time, it is also valuable paper that provides basic information on limb regeneration of salamander.

However, I require some responses to these questions and comments, before acceptance to Nature Communications.

Major comments

1. Each of the two studies (ossification patterns of the bones, and clonal analysis in finger cartilage) is valuable. However, the relationship between each study is unclear. Can the phalange formation mechanism explain the regenerative capacity of the entire arm, including the humerus?

- We agree with Reviewer #2 that making sure that the phalange development and humerus (or ulna and radius) development/regeneration are consistent in terms of cellular mechanisms. To ensure this is the case, we relied on clonal analysis in different cartilages, which allowed us to estimate the proportions and locations of longitudinal and transversal cell divisions. The analysis of sections via humerus and other long bones/cartilages (ulna and radius) (Fig. 4, please navigate to new Supplementary Information File 2 with more examples of transverse clonal patterns, cell flattening and repositioning in Brainbow *Ambystoma mexicanum*) showed the consistent spatial arrangements of chondrocytes – transversal clones in all analysed long bones, flattening re-arranging clones, hypertrophic zones etc. These analyses were done in two species, multiple stages and the outcome was analysed after complete regeneration was achieved, as the visualisation of the clones for analysis requires tissue processing. Our observations of cells repositioning and changing orientation of cell division *in vivo* in the phalange are consistent with the positional distribution of clones in the fully regenerated bigger long bones, both in *Ambystoma* (Fig. 4C, Fig. S5A, Fig. S6B-D) and *Pleurodeles* (Fig. 4A, B, Fig. S5B-H, Fig. S6A,E), in development as well as regeneration (Supplemental Information File 2). Both the phalange and the humerus are long bones, which is why they fit together in the present study. The benefit of the results obtained in the phalange is that we could trace individual cells in live imaging *in vivo* during a regenerative period (Fig. 3), which would be impossible to do in the much larger ulna, radius or humerus.

2. The micro-CT is a new technology for regeneration research in salamanders. The authors need to show how accurately distinguish the areas of the cartilage and the bones by analysing the micro-CT data

- We understand the reviewer's concern, and we would like to resolve it by saying that we have long-lasting expertise with tissue contrasting, micro-computed tomography and image segmentation, specifically, in the area of stiff tissues – cartilage and bone, and, even more, in salamanders: Please navigate to the following salamander cartilage and bone publications:

1. "Living in darkness: Exploring adaptation of *Proteus anguinus* in 3 dimensions by X-ray imaging."

Tesařová M, Mancini L, Mauri E, Aljančič G, Năpăruș-Aljančič M, Kostanjšek R, Bizjak Mali L, Zikmund T, Kaucká M, Papi F, Goyens J, Bouchnita A, Hellander A, Adameyko I, Kaiser J. *Gigascience*. 2022

<https://www.ncbi.nlm.nih.gov/pmc/articles/PMC8982192/>

2. "A quantitative analysis of 3D-cell distribution in regenerating muscle-skeletal system with synchrotron X-ray computed microtomography." Tesařová M, Mancini L, Simon A, Adameyko I, Kaucká M, Elewa A, Lanzafame G, Zhang Y, Kalasová D, Szarowská B, Zikmund T, Novotná M, Kaiser J. *Sci Rep*. 2018

<https://www.nature.com/articles/s41598-018-32459-2>

Please, see the other publications of our team that in detail describe the methodology and show the preciseness of segmentations, including validation by immunohistochemistry and histology:

- <https://elifesciences.org/articles/25902>
- <https://elifesciences.org/articles/34465>
- <https://academic.oup.com/gigascience/article/10/3/giab012/6156288>
- <https://iopscience.iop.org/article/10.1088/1748-0221/11/03/C03006/pdf>

The details of the segmentation have been updated in the Methods section. In order to help the reader to visualize the process, some steps of the methods are shown in Fig. 1A, Fig. 2B. Of note, we compared the results of our segmentation with Alcian Blue (stains cartilage, see Fig. 1E-H, Fig. S3B) and von Kossa (stains bone, see Fig. 1E-H, Fig. 4C, Fig. S3B) staining during the entire work and preparation of this manuscript to ensure consistency.

3. Although the sample number of regenerative limbs is small (maybe 3?), are the observed phenomena common for regenerative process? Can you quantify the area of ossification?

- Yes, absolutely! In the revised version of the manuscript, we have included the quantifications of the areas of ossification from 3D reconstructions for the most informative samples: postmetamorphic *Pleurodeles* (Fig. 1C) and pre-metamorphic *Ambystoma* (Fig. S4D), please, see new panels.

4. EdU is not sufficient for the lineage trace (Fig. 6). Is it difficult to combine NucCyt and EdU labeling? Alternatively, show the M-phase figure of Edu-positive cells. If the direction of cell division is known from the direction of the mitotic spindle, it will help the reader's understanding.

- We agree, and to address the comment we have incorporated a new inset in Fig. 4A showing positive EdU+ doublets, including typically occurring mitotic spindles (pointed metaphase plate), both oriented transversally to the length-axis of the skeletal element. Next, we performed the requested experiment by combining EdU pulse-chase and clonal labelling. The outcome of these combined experiments strongly supports the absence of cartilage stem cell niches in the epiphysis of long bones and supports the changes in the orientation of cell division/cellular repositioning (see new Fig. S7).

Minor comments

1. Maybe, it is a typo. Fig. 1A, middle bottom "Adult stage 56".

- We are grateful for this comment, but this is not a typo. Stage 56 corresponds to post-metamorphic *Pleurodeles* salamanders, including both juveniles and adults. To avoid confusion, in the new Fig. 1 we have changed the "Adult stage 56" to "Stage 56, adult)".

2. In Fig. 2A, "dpa" is displayed above the un-regenerating control. It is a little bit confusing as these limbs did not be amputated.

- The unamputated limbs are contralateral control limbs of unilateral amputations, which is why the "dpa" were displayed above. To avoid confusion, we have moved the "dpa" so the time points are displayed above the regenerating limbs.

3. Also, in Fig. 2A, only "regenerating limb 10 dpa" has been expanded. It is still confusing, despite the authors show it with a scale bar.

- We have considered reducing the "regenerating limb 10 dpa" so all the time points are displayed in comparable sizes. However, we decided to maintain the original outline as the suggested change would severely decrease the visibility of such an important timepoint (making it unacceptably tiny for the journal figure).

4. How do you distinguish the plane of amputation? I guess, it is shown as the bone misalignment like regenerating limbs in Fig. 2A. If so, it looks strange in the regenerating limb (st 55) in Fig. 2B right. Also, correct the red-dotted line extending to the palm.

- The plane of the amputation can be distinguished in the uCT-scans by the presence of the original bone and the general shape of the skeletal element. We appreciate the comment, as it made us realise that some of the amputation planes in Fig. 2A were slightly misplaced (we have corrected it now). The planes depicted in figure 2B are not amputation planes but the level of the optical sections shown below. We have modified the colour of the amputation plane to cyan in Fig. 2A, so they are not confused with optical sections in Fig. 2B and adjusted the description in the figure legend.

5. Fig.4 and Fig. 5 are look very busy. I recommend organizing the layout of the panels and how to add captions. Especially, the direction of the rose diagram is difficult to read.

- We agree and have re-organized all the figures in the latest version of the manuscript. Specifically, we have changed the representation of the rose diagrams to violin plots to make them consistent and easier to read.

Reviewer #3 (Remarks to the Author):

The paper analyses the way bones are formed during salamander limb regeneration. By μ CT the authors find that -in contrast to mammalian limb development - the regenerating skeleton is formed by chondrocytes, which are only replaced by bone after the final size of the regenerating limb is reached. Using genetic, multicolor fate mapping tools (brainbow) they show that after condensation chondrocytes align horizontal to the long axis of the skeletal element mimicking basicranial synchondrosis development more than that of mammalian long bones. They further investigate Sox9/Col2 and Ihh/PthrP expression, which are expressed in the expected regions of the cartilage anlagen. The differences to the embryonic and postnatal differentiation of mammalian long bones are very interesting observations and of importance towards an understanding of skeleton regeneration.

However, while the paper nicely describes and analyses bone formation by μ CT and cell division planes in genetically labeled cells the experiments are quite redundant. While the investigation of two species at pre- and post-metamorphic stages each clearly supports the findings, it makes reading quite repetitive. Similarly, the figures are quite repetitive analysing ossification by μ CT: fig. 1-3 & fig. S1, S4 division plane fig. 4-6 and fig. S3). It might make more sense to restrict the analysis in the paper to one species and include the other data in the supplements.

- We would like to thank the reviewer for revising our manuscript and for the valuable suggestions. We appreciate the comment regarding how investigating two species and stages strengthen the findings. In this revised version, we included further experiments that support the claims, as suggested by all reviewers. We also agree and understand that showing redundant or similar results is tiring for the reader. Thus, following this reviewer's suggestion, we have re-arranged the figures moving similar and confirmatory results into supplementary (for instance, *Ambystoma* microCT data and many more). We have also modified the text to make the reading more appealing.

In fig. 4b and 5c the authors investigate developing bones. These seem to undergo transverse division planes similar to the regenerates indicating that the plane of division is characteristic for Salamanders rather than regeneration.

- We absolutely agree with the reviewer, as both longitudinal and then transversal cell divisions are characteristics of both development and regeneration in salamanders. This is now clarified and presented as a new summary Figure 7, which we added during this revision. Also, this general principle is evident from the included analysis of salamander developing bones in this version of the manuscript Fig. 3, Fig. 4, Fig. S6. Similar to regeneration, we observed changes in the orientation of cell division and repositioning of cells in development, despite the changes being slightly less marked, with most doublets oriented in an oblique way during homeostatic development Fig. S6B-C. We can therefore claim that the change of the division plane is, as the reviewer pointed out, a general characteristic for salamanders. However, there is still a difference between development and regeneration, especially when it comes to the timing and proportions of the longitudinal cell division phase and transversal cell division phase, which overall results in bulky bones in regeneration (articulated in revised Fig. 1 and Fig. 2).

We also provided a wealth of representative images of clonal arrangements in *Ambystoma*, both in development and regeneration. Please, navigate to the new Supplemental Information File 2 (extended multi-page pdf).

Some information about the stages analysed and how they relate to mouse development would also be important. Fig. 5 CD show sections through synchondroses and long bones of mice. Why are the stages analysed different (pre and postnatal) and how do they relate to the investigated Salamander stages?

- We agree and added the description and stages of the development of salamander limb skeletal elements throughout the text and in figure legends. For the case of Figure 5C/D (now Fig. 4), the mentioned stages correspond to the tamoxifen administration at E14.5 and the tissue analysis at P30. At E14.5, the mouse limb is fully patterned, but it will grow considerably in size until P30. Also, at E14.5, the limb skeletal elements are made of stratified cartilage in mouse embryos. This is similar to the situation found in *Pleurodeles* larval stages 45-52. In this revision, we clarify this in the updated figure legends of Fig. 4:

"The recombination was induced at E14.5 in Col2^{CreERT2}/R26^{Confetti} mouse embryos, where while the mouse limb is fully patterned, the skeletal elements are made of stratified cartilage. The chondrogenic clones were analysed at P30."

The basisphenoid cartilage was analysed at another developmental stage (E17.5) because that is the time for the transition into ossification in basicranial synchondroses. This is also reflected now in the corresponding figure legend to Fig. 4:

"The recombination was induced at E12.5 in Sox10^{CreERT2}/R26^{Confetti} mouse embryos, and the chondrogenic clones were analysed at E17.5. The basicranial cartilage undergoes ossification at the E17.5 and allows for observing the cell dynamics in synchondroses."

Additionally, we have added pictograms into the Fig. 4 to highlight the comparison between species and marked the area of skeletal elements analyzed and presented in the panels.

Fig. 8; fig. S5 (now S6): The analysis of Sox9/Col2/Ihh/Gli/PthrP expression is interesting, but it is not clear what the authors want to conclude? That the pathways act in the same way? How does that contribute to or explain the transverse division planes?

- We agree with the reviewer that there was an issue with the early version of the manuscript, and we improved the explanations of our findings to cope with this. In the first submission, we found the dynamics in the expression patterns of the PTHrP/Ihh signalling loop, but we likely did not manage to convey the message. That is why we now added the schematic view that summarises where each of the genes was found; please, see the new summary Figure 7, where we explained the meaning of our findings in terms of developmental vs regenerative differences in the Results section:

"... we showed that PTHrP/Ihh loop has a dynamic expression pattern during regeneration, in contrast to the gradient maintained during developmental growth in salamanders. The continuous endochondral ossification during development shows a constant pattern of moderate expression of

PTHrP in the proliferative zone, intermediate Gli1, and Ihh expressed in the chondrocytic cells (Fig. S3C). In contrast, our analyses revealed that PTHrP/Ihh loop has a dynamic expression domains during regeneration that vary in size and shape compared to their contralateral control limbs (Fig. 6, Fig. S8). First, cartilage condensation is characterised by a switch to a radial polarity pattern of Ihh-Gli1-PTHrP from the core to the periphery (Fig. 6B, C). Second, Ihh+ pre-hypertrophic chondrocyte domain expands and forms the callus wrapping the amputated bone, while the growth continues distally, characterised by Gli1+ columnar chondrocytes and PTHrP+ articular chondrocytes (Fig. 6D, E). Third, the gradient observed in development is restored but with extended expression domains compared to the unamputated controls (Fig. 6F). Finally, we found Gli1 and PTHrP double-positive signal in enlarged chondrocyte cells located proximally to the flat cells (Fig. 6G4), which are likely hypertrophic chondrocytes (both in developing and regenerating limbs). We also found periskeletal cells positive for Gli1 and PTHrP, which could be contributing to the developing as well as the regenerated cartilage (Fig. 6C⁴, E3, F6, G1, S8B⁴).”

The expression is difficult to see, especially at a later stage. It would be helpful to display the original (not combined) pictures at least as supplement, to better evaluate the expression of single genes.

- The images obtained by RNAscope (Ihh/Gli/PthrP) are hard to evaluate at low magnification in regular figures because of the small dot size corresponding to an mRNA signal. In order to respond to this comment, we now included them in supplementary materials to show the spatial context in which different cells were found (please find them in updated Fig. 6, Fig. S3C, Fig. S8). We provided high-resolution images of individual channels as Supplemental Information Files 1, 3, and 4. At the same time, the new magnified insets of regions of interest show representative cells of each part of the regenerating cartilage elements and, in our opinion, are the best way to show the results in updated Fig. 6, Fig. S3C, Fig. S8.

Are Sox9 and Col2 exclusively expressed (fig. 8; fig. S2, S5)? This seems to be surprising.

- Sox9 and Col2a1 are co-expressed in both the developing and the regenerating limbs at early stages (previously Fig. 8B, D; S3B, C, D – now Fig. 6B,D and Fig. S2B). Although surprising, we also detected non-overlapping expression in the chondrocytes that seem to be becoming hypertrophic (Fig. 6D; Fig. S2E) together with a dramatic decrease of expression at later stages (Fig. S2F). The much lower levels of Sox9 can explain this apparent uncoupling of Sox9 and Col2a1 signal in hypertrophic chondrocytes, which sometimes fall under the detection capacity.

Reviewer #4 (Remarks to the Author):

The presented article shows comparative data on the ossification process in the context of limb development vs. limb regeneration in salamander models. The findings here demonstrate marked differences between limb development and limb regeneration that help us to understand one of the long-standing questions as to whether regeneration is a recapitulation of the developmental program and whether the cellular processes involved in the morphogenesis of regenerated tissues mirror the events of embryonic development. On the other hand, the findings presented here show interestingly the conservation of growth and ossification mechanisms between regenerating limbs and the development of craniofacial structures such as synchondroses at the cranial base.

- We cordially thank Reviewer 4 for the interest in our article. We hope that the revised version became better and more comprehensive.

Axolotl Bone Development after amputation

HCR Investigation

HCR Panel

Shh
Col2a1
Bmp2
Bmp5
Bmp7
Msx1
Msx2
Fgf8
Fgf10

Time-frame: 3w, 5w, 6w weeks after amputation

Controls: Stage 3.5cm, 6cm, 12cm. Corresponding to pre-calcifying, intermediate, and calcified.

Gene List: Based on:

STEM CELLS AND REGENERATION RESEARCH ARTICLE

BMP signaling is essential for sustaining proximo-distal progression in regenerating axolotl limbs

Etienne Vincent, Eric Villiard, Fadi Sader, Sabin Dhakal, Benjamin H. Kwok and Stéphane Roy^{1,2,*}

Conclusions

Unfortunately many probes did not give useful results, either due to being involved at earlier budding stages, or that the HCR probes were not able to pick up signal.

Msx2 expression increases after amputation, with obvious expression in the newly growing bone area, but not in the old bone.

Msx2 can also be seen in control animals, not in the arm bones but in the hand bones. This could be stage-dependent though, and we might miss the expression window in the arm bones.

Full Resolution raw image files are available at:

<https://figshare.com/s/5912f0473be226dbcf80>

Msx2 Results

Control, 3.5cm stage

Msx2
Col2a1
DAPI

Msx2 Results

3 weeks post amputation

Msx2 Results – Later Stages

Control – 6cm: No Msx2 detected.

Msx2
Col2a1
DAPI

5w Post amputation: Msx2 expressed in new bone, but no longer nuclear.

Msx2
Col2a1
DAPI

Unsuccessful probes

3w post-amputation

Bmp5
Bmp2
DAPI

Msx1
Bmp7
DAPI

Unsuccessful probes

6 cm control

Bmp5
Bmp2

Msx1
Bmp7

Blank
Fgf10

Shh
Col2a1

Msx2
Fgf8

REVIEWERS' COMMENTS

Reviewer #1 (Remarks to the Author):

General comments: I believe the authors have done an admirable job of addressing reviewer comments and applaud their effort. My only general concern is that the MS could be shortened considerably and still make the specific scientific points desired by the authors.

Specific comments:

1. Results: the set of experiments that used inhibitors to signaling pathways should be eliminated as the information was mostly inconclusive or no effect and the graphs in (A) have only 2 points that are connected without really knowing if they should be a straight line. The graphs in D and E show no significant differences. For B and C, data show differences for specific bones which is confusing.
2. Discussion: Much of the discussion is also stated in the Results. It would be preferable to remove all but the experiments from the results and build the story in the discussion.

Reviewer #2 (Remarks to the Author):

They have added more data in this revise. I respect their efforts. May be somewhat inadequate in terms of directly revealing limb regeneration at the molecular level. However, they successfully took on the challenge of obtaining new answers using an experimental system of the salamanders with underdeveloped molecular tools. Their results will serve as the basis for future development of salamander regeneration research.

In addition, they have responded well to my questions and requests. However, please continue to consider the following

1. With regard to the question of whether the phalanx formation mechanism can explain the regenerative capacity of the entire arm, including the humerus, I agree with the authors' answer. I would like to see this response reflected (or added) to the text.
2. I was convinced that the authors are professionals in identifying cartilage. However, for the readers who share my concerns, I would like you to provide in the text the citations on which you based your discernment.
3. If you want to compare Figures 1F and 1E, you should add a scale bar to 1F, and a scale bar to 1E.

Reviewer #3 (Remarks to the Author):

The authors have addressed most of my concerns.

The findings of the paper are of high interest identifying not only differences between the organization of chondrocytes during limb development and regeneration, but also between salamander and mammalian bone development.

I have a few further comments:

Fig. 3: it would be helpful to include the time intervals of the analysis in the figure legends or figure
Where do the lower panels 'formation of clonal columns' and 'cell repositioning' come from? Are these magnifications?

A/B: Demarcating the regions 'mesenchymal' and 'maturing' in the figure would help the reader.

Fig. 4 legend:

Line 1013: do you mean S6h instead of 6gH?

Legend D and E should be swapped

Lines 217-258 slightly redundant text,

Lines 345-361 do not represent new results

Both paragraphs summarize and discuss the results extensively. It might be helpful to shorten these paragraphs especially as the discussion also refers to them.

Fig. S9: I do not understand the figure legend. What is compared, what are the different parameters for significance? What does **** zeugopodial elements mean. Which parameters are n.s. n.s. appears in the text but not in the figure.

The method e.g. concentration and timing of the individual treatments also lacking.

REVIEWERS' COMMENTS

Reviewer #1 (Remarks to the Author):

General comments: I believe the authors have done an admirable job of addressing reviewer comments and applaud their effort. My only general concern is that the MS could be shortened considerably and still make the specific scientific points desired by the authors.

- We would like to cordially thank the reviewer for this positive feedback, appreciation of our efforts and the suggestions. We have followed the reviewer's advice and removed parts of the text and inconclusive data that were generated for the previous manuscript revision (Figure S9 and related descriptions). We also removed some redundant parts from the Results and made Discussion more focused.

Specific comments:

1. Results: the set of experiments that used inhibitors to signaling pathways should be eliminated as the information was mostly inconclusive or no effect and the graphs in (A) have only 2 points that are connected without really knowing if they should be a straight line. The graphs in D and E show no significant differences. For B and C, data show differences for specific bones which is confusing.

- We agree with the reviewer, and we removed the Supplemental Figure 9 together with related description and discussion of these results.

2. Discussion: Much of the discussion is also stated in the Results. It would be preferable to remove all but the experiments from the results and build the story in the discussion.

- We thank the reviewer for pointing out the information redundancy. We have removed the repetitive parts from the Results and focused on them in the Discussion. Please see the changes in the main text file highlighted in yellow. We really hope that the revised structure of the text is easy for reading and informative in all sections.

Reviewer #2 (Remarks to the Author):

They have added more data in this revise. I respect their efforts. May be somewhat inadequate in terms of directly revealing limb regeneration at the molecular level. However, they successfully took on the challenge of obtaining new answers using an experimental system of the salamanders with underdeveloped molecular tools. Their results will serve as the basis for future development of salamander regeneration research. In addition, they have responded well to my questions and requests. However, please continue to consider the following

1. With regard to the question of whether the phalanx formation mechanism can explain the regenerative capacity of the entire arm, including the humerus, I agree with the authors' answer. I would like to see this response reflected (or added) to the text.

- We agree with the reviewer that it is important to highlight the shared cellular dynamics found in phalanges, ulna, radius and humerus in development and regeneration. We have added this information to the main text:

"We could not perform a live imaging on clonally traced ulna, radius or humerus due to their thickness, therefore, we analyzed clonally traced tissue sections of developing and regenerated humerus, ulna and radius to show the consistent spatial arrangements of chondrocyte clones in all skeletal elements (Fig. 4; navigate to Supplemental Information File 2 for extended clonal tracing results). We confirmed that development and regeneration of phalange, humerus, ulna and radius employ similar cellular dynamics during growth and shaping."

2. I was convinced that the authors are professionals in identifying cartilage. However, for the readers who share my concerns, I would like you to provide in the text the citations on which you based your discernment.

- We agree, and we have added these citations into the updated text:

"The identification of the border between soft tissues, cartilage and bone was done according to our previously published approach allowing reliable segmentation of soft and stiff tissues in vertebrates, and specifically in salamander species^{23,28-31}."

3. If you want to compare Figures 1F and 1E, you should add a scale bar to 1F, and a scale bar to 1E.

- Scale bars are now added to Figure 1E and 1F for better comparison.

Reviewer #3 (Remarks to the Author):

The authors have addressed most of my concerns. The findings of the paper are of high interest identifying not only differences between the organization of chondrocytes during limb development and regeneration, but also between salamander and mammalian bone development.

- We thank the reviewer for the appreciation of our findings and the valuable feedback. We have followed the reviewer's advice and incorporated all changes. We are very grateful for the reviewer's time and efforts spent on our manuscript.

I have a few further comments:

Fig. 3: it would be helpful to include the time intervals of the analysis in the figure legends or figure

- We absolutely agree, and we have added the time points into each picture.

Where do the lower panels 'formation of clonal columns' and 'cell repositioning' come from? Are these magnifications?

- The image at the bottom of this panel (demonstrating the formation of clonal columns) is the magnification of an area in 18 d.p.a. sample from panel B; cell repositioning is presented as magnified individual planes from z-stack files from the same live-imaging series showing the timecourse of phalange development. This is now stated in the corresponding figure legend.

A/B: Demarcating the regions 'mesenchymal' and 'maturing' in the figure would help the reader.

- We thank the reviewer for this suggestion, we have marked the regions in both panels and added respective description into the figure legend.

Fig. 4 legend: Line 1013: do you mean S6h instead of 6gH? Legend D and E should be swapped

- We thank the reviewer for bringing these two mistakes to our attention, we have corrected them, and the legends are swapped.

Lines 217-258 slightly redundant text,

- We reorganized the manuscript text to be more focused and less redundant.

Lines 345-361 do not represent new results

- We understand that previous lines 345-361 were a bit repetitive, as these lines are summarizing the main findings described in the previous paragraphs (lines 295-335 in the previous version). Due to the amount and quality of the data, we consider it is important to describe the results both in detail for each of the regenerating timepoints analyzed (291-327 in the last version) as well as summarizing the mayor events observed (328-343 in the last version). In any case, we tried to improve the text and remove repetitive phrases for better readability.

Both paragraphs summarize and discuss the results extensively. It might be helpful to shorten these paragraphs especially as the discussion also refers to them.

- We shortened the paragraphs (especially in the Results) where it was possible without damaging the logic of narrative.

Fig. S9: I do not understand the figure legend. What is compared, what are the different parameters for significance? What does **** zeugopodial elements mean. Which parameters are n.s. n.s. appears in the text but not in the figure. The method e.g. concentration and timing of the individual treatments also lacking.

- We are very thankful for this question, but based on the new request of the Reviewer #1, we have removed these data and the entire Supplemental Figure 9, respective figure legend and description in the text, as the Reviewer 1 pointed out that these data are inconclusive in regard to the question of the paper.